# Hydro-meteorological reconstruction and geomorphological impact assessment of the October, 2018 catastrophic flash flood at Sant Llorenç, Mallorca (Spain)

Jorge Lorenzo-Lacruz[1], Arnau Amengual[2], Celso Garcia[1], Enrique Morán-Tejeda[1], Víctor Homar[2], Aina Maimó-Far[2], Alejandro Hermoso[2], Climent Ramis[2], Romualdo Romero[2]

[1]Department of Geography, University of the Balearic Islands, Palma, 07122, Spain
[2]Department of Physics, University of the Balearic Islands, Palma, 07122, Spain

*Correspondence to*: Jorge Lorenzo-Lacruz (j.lorenzo@uib.es)

**Abstract.** An extraordinary convective rainfall event, unforeseen by most numerical weather prediction models, generated a devastating flash flood (305 m$_3$s$_{-1}$) in the town of Sant Llorenç des Cardassar, Mallorca on October 9, 2018. Four people died inside this village, while casualties were up to 13 over the entire affected area. This extreme event has been reconstructed by implementing an integrated flash flood modelling approach in the Ses Planes catchment up to Sant Llorenç (23.4 km²), based on three components: (i) generation of radar-derived precipitation estimates; (ii) modelling of accurate discharge hydrographs yielded by the catchment (using FEST and KLEM models); and (iii) hydraulic simulation of the event and mapping of affected areas (using HEC-RAS). Radar-derived rainfall estimates show very high agreement with rain gauge data ($R^2 = 0.98$). Modelled flooding extent is in close agreement with the observed extension by Copernicus Emergency Management Service, based on Sentinel-1 imagery, and both far exceed the extension for a 500-year return period flood. Hydraulic simulation revealed that water reached a depth of 3 m at some points, and modelled water depths highly correlate ($R^2 = 0.91$) with *in-situ* after-event measurements. The October 9 flash flood eroded and transported woody and abundant sediment debris, changing channel geomorphology. Water velocity greatly increased at bridge locations crossing the river channel, especially at those closer to Sant Llorenç town centre. This study highlights how the very low predictability of this type of extreme convective rainfall events and the very short hydrological response times typical of small Mediterranean catchments continue to challenge the implementation of early warning systems, which effectively reduce people's exposure to flash flood risk in the region.

## 1. Introduction

The European Union member states reported 3,198 floods occurred between 1980-2015 (EU Floods Directive 2007/60/EC) (http://rod.eionet.europa.eu/obligations/601; https://www.eea.europa.eu/data-and-maps/data/european-past-floods). These floods were responsible for 4,806 fatalities and an approximate total damage cost of 115,113 million euros. Spain reported

more than 20 floods per 10,000 km2, one of the highest in Europe, with 652 fatalities (European Environment Agency, 2016). In 2018, flash floods killed 152 people across Europe, parts of northern Africa and the Middle East. The deadliest flash floods occurred in the Mediterranean area (Mallorca, Tunisia, Veneto, Sicilia, and Jordan), including 5 events with 70 fatalities, concentrated in almost one month: 9, 17 and 29th October; 3 and 9th November (European Severe Weather Database; https://www.eswd.eu/).

Flash floods are known as high-intensity precipitation episodes with a convective origin that fall over small catchment headwaters, generating a fast hydrological response characterised by a sudden flow event with steep rising and falling limbs and short lag times (i.e., Marchi et al., 2010; Borga et al., 2014). In the western Mediterranean region, flash floods occur mostly in the autumn (Llasat et al., 2010) and generally they are more intense than in continental areas (Gaume et al., 2009). The relatively high sea surface temperature increases the convective available potential energy of the overlying moist air masses

through sensible and latent heat flux exchanges. Together with the intrusion of cold air aloft and the presence of mesoscale vertical forcing mechnisms, the complex orography and acute land-sea contrasts, promote the lifting of low-level unstable air, favouring the triggering of deep moist convection (Romero et al., 1998; Martínez et al., 2008; Cohuet et al., 2011). Despite some important flash floods are produced by heavy precipitations that last less than 1 or 2 hours (Martín-Vide and Llasat, 2018), high precipitation rates can remain during several hours over individual catchments. This persistence is often associated

with prominent orography or quasi-stationary convergence lines that anchor convection (Kolios and Feidas, 2010; Amengual et al., 2015).

In the Spanish Mediterranean region, many small- to medium-sized catchments are steep and highly urbanized, and flood-prone areas are densely occupied. Many of these rivers are ephemeral, with short hydrological response times, and they are dominated by extreme events of low frequency and high magnitude. All these hydrological characteristics make flash flood

forecasting and warning a huge challenge for flood risk management (Marchi et al., 2010) and, more specifically, create challenges for flood risk perception, awareness and communication strategy (i.e., Bodoque et al., 2019).

The present study tackles a forensic reconstruction of the catastrophic flash flood that occurred on October 9, 2018 in the ephemeral river of Ses Planes, which crosses the town of Sant Llorenç des Cardassar, located east of the island of Mallorca (Fig. 1). An extreme rainstorm triggered a large magnitude flash flood that eroded and transported woody debris and sediment

from the headwaters and along the fluvial valley, killing 13 people (4 in the town of Sant Llorenç) and damaging more than 300 dwellings, 30 stores and 324 vehicles (https://sinobas.aemet.es/index.php?pag=detal&rep=1221). The total damage cost of the event was estimated at 91 million euros (Govern de les Illes Balears, 2018).

The methodology used for the reconstruction of the event was organized on three main stages: (i) 10-minutes precipitation has been derived from radar reflectivity observations; (ii) two distinct fully-distributed hydrological models have been

implemented to accurately simulate the discharge hydrograph and; (iii) a hydraulic simulation has been performed to map the affected areas, including flooding extent and timing, water depth and flow velocity. Some of the geomorphological impacts on the main channel have been also assessed by using very high-resolution orthophotographs and digital elevations models for comparison of pre- and post-flooding conditions.

This deadly flash flood had an important social impact on the Balearic Islands and in the entire country (the urban flooded area was already included in the Spanish national flood risk maps portal). The operative mesoscale meteorological model by the Spanish meteorological agency (AEMET) failed at reproducing the convective clusters which triggered the intense rainfall episode over the area. Consequently, AEMET shifted from orange to red alert for intense rainfall only when the flood peak was nearly upon the town. This extreme flood event has shaken the foundations of flood risk perception for the people of the Balearic Islands. How to face this challenge for hydro-meteorological modellers and forecasters, civil protection managers and policy makers is discussed in this paper.

## 2. Case study

### 2.1 Catchment description

The Ses Planes torrent is a small ephemeral stream located on the eastern part of the island of Mallorca (Fig. 1). The catchment has a drainage area of 23.4 km2 at the entrance to the town of Sant Llorenç des Cardassar (8405 inhabitants). It is located in the Llevant Ranges, a Neogene horst structure composed mainly of carbonate deposits that have an Alpine internal compressional structure (Sabat et al., 1988). The topographic relief and the steepness of the slopes are important (watershed average slope of 18%); over a few kilometres, the elevation increases from less than 100 m to small mountains close to 500 m. The catchment has a well-developed drainage network, with a circular (fan) shape on the headwaters and an elongated bottleneck shape before the entrance of the town (Fig. 1). Forested areas are reduced and vegetation density is generally low, due to the prominence of sclerophyllous vegetation and rainfed crops, conditioned by thin soils. The Ses Planes catchment exhibits fast hydrological responses to heavy precipitation, resulting in fast Hortonian flows during intense rainfall episodes (Estrany et al., 2010). However, high infiltration rates are also present because of the persistence of low soil moisture content and the existence of underlying karstic and dolomitic fractured bedrock that promotes deep percolation. Therefore, this catchment features a highly nonlinear hydrological response to heavy precipitation and large rainfall amounts.

The Ses Planes torrent has been affected by some severe flood events over the last half century: 12/10/1973, 3/09/1982, 25/10/1985 and 06/09/1989 (Grimalt and Rodríguez-Perea, 1989). The latter was especially important; 156 mm of precipitation recorded in two hours generated a flash flood that affected some areas of the town and led local authorities to undertake the artificial channelization of the Ses Planes torrent where it crosses the Sant Llorenç urban area.

This small Mediterranean catchment is not gauged by any local water agency. Nevertheless, the region is monitored by a Doppler C-band weather radar of the AEMET network located 60 km away from the catchment (Fig. 1). Within this region, raw precipitation is recorded by four automatic rain-gauge stations with a temporal resolution of 10 minutes. Daily rainfall amounts are also recorded in 17 additional rain gauges, all of them located outside the Ses Planes watershed limits (Fig. 3a). All the pluviometric and thermometric weather stations belong to the AEMET observational network.

## 2.2 Synoptic situation

The large-scale meteorological pattern responsible for the catastrophic flash flood event was very well defined, according to the circulation systems seen at 500 hPa and at surface on that day (Fig. 2a). The 500 hPa analysis shows a deep trough that covered the North Atlantic, with a strong gradient of geopotential height at mid-latitudes that induced a strong south-westerly flow towards western and northern Europe. The high geopotential centre located over the northern sector of continental Europe favoured this large-scale flow. In contrast, the geopotential gradient was much weaker to the south, covering part of Morocco, Algeria and the western Mediterranean.

In direct connection with the genesis of stormy weather over the Balearic Islands, a cold cut-off low of small dimensions (diameter of about 600 km) was present at the same mid-tropospheric level over the Iberian Peninsula and its Mediterranean coast. This disturbance is well marked in the synoptic map and implies a southerly flow over the western Mediterranean, especially along its eastern flank. The depression was associated with a positive vorticity anomaly to the south-southwest that was advected by the mid-level winds towards the position of the Balearics and the north-eastern lands of the Iberian Peninsula. This classical synoptic pattern, when combined with a low-level moisture feeding mechanism and convective or latent instability in the atmospheric column, is known to be conducive to heavy precipitation in Mediterranean Spain (Doswell et al., 1998; Romero et al., 1999).

Surface isobars (white contours in Fig. 2a) reveal, in addition to the low-pressure centres of the North Atlantic, a powerful anticyclone over Eurasia that covered most of the Mediterranean and determined a general flow from the southeast in the western Mediterranean region. This maritime flow was most pronounced towards the Balearic Islands and the Spanish Mediterranean coast, owing to the presence of a shallow and weak depression extending from the southern half of the Iberian Peninsula towards the Atlantic coast of Africa. The synoptic situation at 850 hPa (not shown) was very similar to the surface circulation, displaying a marked warm air advection from the southeast towards the Balearic Islands. This thermal advection is confirmed by the presence of veering winds between low tropospheric levels and 500 hPa. The humidity records taken at the main meteorological stations of the western Mediterranean (not shown) reveal that the air advected towards the Balearic Islands also had a high moisture content, thus helping to develop latent instability conditions over the western Mediterranean. In addition, the persistence of this warm and moist flow during October 9 was most likely instrumental in sustaining the convective systems once initiated.

Figure 2b shows the IR satellite image at 1700 h UTC (1900 h Local Time; hereafter LT), that is, at the time the torrential rainfall was occurring over the study area (see Figs. 3 and 4). This image clearly shows the cloud structure associated with the upper-level cold low, a plume extending from North Africa along the western Mediterranean with embedded intense convective nuclei. Several convective cores stand out with very cold tops to the south and above the Balearic Islands, illustrating the powerful character and high depth of the convective cells. During previous hours (images not shown), several convective clusters could already be identified to the south of the Balearic archipelago. These systems then moved northwards, affecting north of Mallorca and west of Menorca during the afternoon and evening hours. Figure 4 show how the precipitation

evolved in response over the affected area, following a southwest to northeast direction. These extreme accumulations from a
sequence of mature storms repeatedly affected the northeast of Mallorca.

The fact that the convective systems formed east of the small cut-off low identified at 500 hPa (see Fig. 2a and 2b) is consistent with the dynamical influence attributed to these isolated cold disturbances: the eastern flank is the zone with the most favoured upward motion across a deep column and, therefore, the preferential area for an organized and intense release of convective instability. Since these convective nuclei developed over the sea, the trigger mechanisms most likely consisted of some sort of
low-level convergence line defined by mesoscale factors ahead of the above-mentioned southeasterly current.

## 3. Methods

### 3.1 Atmospheric modelling and convective precipitation predictability

An initial numerical exploratory study was performed after it was ascertained that no operational system forecasted
precipitation rates over eastern Mallorca anywhere near the recorded rainfall rates in their operational cycles (Figs. 2c and 2d). In particular, the operational deterministic European Centre for Medium-Range Weather Forecasts (ECMWF) model barely produced 20 mm of accumulated precipitation over northeast Mallorca during the afternoon of 9 October, for any forecast cycle. Regarding the operational HARMONIE-AROME model at AEMET, even a more severe underestimation of the amounts was forecasted over the area of interest. These models operate with grid resolutions of a few kilometres. We explored the
potential of sub-kilometre resolutions using the Weather Research and Forecasting (WRF) Model, version 3.9. Specifically, a mesoscale (900 m) and a microscale (100 m) resolution simulations were performed, centred over the Balearic Islands and the Ses Planes basin, respectively, steered by possible hydrological and hydraulic modelling requirements. These numerical simulations are initialized with the ECMWF operational products. The WRF configuration is a standard suite of parameterizations used by Amengual et al. (2017) and also described at https://meteo.uib.es/wrf. To account for uncertainties
in initial conditions of the simulations, and to examine the potential guidance that dynamical downscaling of the 50 members of the ECMWF EPS could provide for this case, the WRF runs were nested in the 0000 h UTC October 9 operational cycle.

### 3.2 Precipitation analysis

The Doppler C-band weather radar located close to the city of Palma performs volume scans with spatial and temporal resolutions of 1 km and 10 minutes, respectively (Fig. 1). On October 9 2018, quantitative rainfall estimations have been
derived from the raw radar volumetric reflectivity scans at 0.5° and 1.5° elevations. Radar blockage has been amended by performing numerical simulations of radar beam propagation over a digital terrain model of the radar domain. These simulations aim at computing the fraction of the pulse volume blocked by the regional orography (Pellarin et al., 2002). Next, reflectivities from the two lowest radar elevation scans have been considered, depending on the percentage of blocked beam at each level. Reflectivity has been converted to rainfall rate by using the WSR-88D convective relationship (Hunter, 1996).

Finally, the hourly radar-based rainfall accumulations have been compared against hourly observations from the automatic pluviometric stations of the AEMET. Biases in radar estimates have been corrected by applying a dynamical adjustment against rain-gauge accumulations (Cole and Moore, 2008).

    Once errors in the hourly radar-derived rainfall estimates have been amended, the 24 h cumulative amounts have been verified against the daily precipitation accumulations recorded at all the pluviometric stations. In addition, exceedance areas for the

total cumulative precipitation according to the rain-gauge and radar observational devices have been compared. Exceedance areas have been computed just over the land mass (i.e., excluding the Mediterranean Sea) of the spatial domain in Figure 3a. This inland region has a whole extension of 618.0 km² and a pluviometric rain gauge density of 0.034 gauges km-2 (i.e., one pluviometric station per 29.4 km²). The spatial distribution of the cumulative precipitation observed by the rain gauge network has been obtained after applying kriging with a linear model for the semivariogram fit. While the weather radar delineates

heavy rainfall areas close to 10 km², the pluviometric network captures regions one order of magnitude larger (~100 km²). Therefore, the actual rain gauge network observed exceedance areas impacted by heavy precipitation up to 200 mm, while radar estimates render amounts up to 350 mm for the October 9, 2018 episode (Fig. 3c).

**3.3 Hydrological modelling**

    The fully-spatially distributed FEST and KLEM hydrological models have been implemented to study the Ses Planes

hydrological response to the 9 October 2018 flash flood. As FEST and KLEM account for a varying complexity in the description of the physical processes and model structure, a better quantification of the associated uncertainties is expected when simulating the peak discharges and flow velocities for this extreme event. In addition, possible model dependencies of the results can be detected. FEST is a continuous physically-based model that accounts for the following processes: infiltration, surface runoff, evapotranspiration, subsurface flow and flow routing (Ravazzani et al., 2016). The hydrological model roots in

computing the soil moisture fluxes by solving the water balance equation at each grid point (i, j):

$$\frac{\partial \theta_{ij}}{\partial t} = \frac{1}{Z_{ij}} \left( P_{ij} - R_{ij} - D_{ij} - ET_{ij} \right) \qquad (1)$$

where $\theta$ is the soil moisture, P is the precipitation rate, R and D are the runoff and drainage fluxes, ET is the evapotranspiration

rate, and Z is the soil depth. Runoff is calculated according to a modified Soil Conservation Service–Curve Number (SCS-CN;USDA, 1986) method extended for continuous simulation. That is, the maximum potential retention S is updated at the beginning of a storm as a linear function of the degree of saturation ($\varepsilon$):

$$S = S_1(1 - \varepsilon) + S_3 \varepsilon \qquad (2)$$


where $S_1$ and $S_3$ are the values of S when soil is dry and wet (i.e., antecedent moisture conditions I and III, respectively). The actual evapotranspiration is calculated as a fraction of the potential rate tuned by the beta function that, in turn, depends on soil moisture content (Montaldo et al., 2003). Potential evapotranspiration is computed according to a modified version of the Hargreaves–Samani equation (Ravazzani et al., 2012). Note that as the actual evapotranspiration does not play any significant role in water abstractions during flash flooding, ET is just computed at daily scale so as to close the water balance equation. The surface and subsurface flow routing is based on the Muskingum-Cunge method (Chow et al., 1988).

As the initial antecedent moisture conditions are set-up to normal in the SCS-CN method, a warm-up period has been performed to achieve an accurate estimation of the initial soil moisture content. To this end, FEST has been forced by using hourly rainfall and daily temperature data over the Ses Planes catchment from 1 August to 9 October 2018 0000 h UTC. During this warm-up period, the state variables evolved from the standard initial values to their appropriate values according to meteorological conditions and parameter values.

KLEM is a simple event-based kinematic hydrological model that accounts for the topographic, soil and vegetation properties. Runoff generating processes are modelled via the SCS-CN approach, while runoff routing is simulated based on a description of the drainage system response (Da Ros and Borga, 1997; Giannoni et al., 2003). That is, runoff propagation relies on the identification of the drainage paths, which includes the characterization of the hillslope and channel networks through a threshold area procedure for basin channelization. Discharge at any location along the river is represented by:

$$Q(t) = \int_A q[t - \tau(x), x] dx \tag{3}$$

where A (km2) denotes the drainage area, q (t,x) indicates discharge at time t and location x, and $\tau(x)$ is defined as:

$$\tau(x) = \frac{L_h(x)}{v_h} + \frac{L_c(x)}{v_c} \tag{4}$$

where $L_h(x)$ is the distance from a generic basin location to the channel network following the steepest descent path, and $L_c(x)$ indicates the length of the subsequent channel path to the basin outlet. That is, runoff propagation roots on two invariant flow velocities along the hillslopes ($v_h$) and channels ($v_c$) of the drainage structure. KLEM also includes a linear conceptual reservoir for base flow modelling, whose structure has been kept invariant. The reservoir input is provided by the rate of infiltration according to the SCS-CN. The initial antecedent soil moisture conditions have been set to dry in the SCS-CN method.

FEST and KLEM has been forced by using the 10-min radar-derived precipitation fields from 9 to 11 October 2018 0000 h UTC so as to accurately simulate the runoff processes during this extreme flood. This time interval has also been considered as the computational time-step for both hydrological models. As high infiltration rates are present over the semi-arid Ses Planes catchment due to the persistence of low soil moisture contents and the existence of underlying karstic and dolomitic fractured

bedrocks that promote deep percolation, this catchment features a highly nonlinear hydrological response to heavy precipitation
and large rainfall amounts. To better cope with the strong nonlinearities of the rainfall-runoff transformation, calibration of both models focussed on assessing the infiltration storativity ($S_0$) and the initial abstraction ratio ($\lambda$) (Table 1). Calibration of $S_0$ permits a proper reproduction of the observed flood water balance when using the CN spatial distribution, while calibration of $\lambda$ copes with the specific lithological features of a watershed (Borga et al., 2007; Amengual et al., 2017).

### 3.3 Hydraulic modelling

The U.S. Army Corps of Engineers' River Analysis System (HEC-RAS) was used for hydraulic modelling and flood mapping, a software that allows 1-D steady and unsteady flow hydraulics calculations (US Army Corps of Engineers, 2016). In this study, HEC-RAS unsteady flow analysis was implemented to obtain flooding water extent and timing, water maximum depth and water maximum velocity during the event. 1-D hydraulic models replicate water movement by solving equations formulated by applying laws of physics. More specifically, HEC-RAS solves equations derived by ensuring mass conservation
(Equation 3) and momentum conservation (Equation 4) between two cross-sections $\Delta x$ apart, which yields the one-dimensional Saint-Venant equations:

$$\frac{\partial Q}{\partial x} + \frac{\partial A}{\partial t} = 0 \qquad\qquad (5)$$

$$\frac{1}{A}\frac{\partial Q}{\partial x} + \frac{1}{A}\frac{\partial(\frac{Q^2}{A})}{\partial x} + g\frac{\partial h}{\partial x} - g(S_0 - S_f) = 0 \qquad\qquad (6)$$

where Q is flow discharge (Q = $uA$, where $u$ is cross-sectional average velocity and $A$ is flow cross-section area), $t$ represents time, $h$ is water depth, $g$ is gravitational acceleration, $S_f$ is frictional slope, and $S_0$ is channel bed slope (US Army Corps of Engineers, 2016). Although equations (5) and (6) have no analytical solution, they can be solved using numerical techniques,
allowing the estimation of $Q$ and $h$ for every cross-section at each time step (Teng et al., 2017).

The HEC-RAS model was chosen because, in terms of predictive analysis, it performs better than more sophisticated models (e.g.: TELEMAC, LISFLOOD-FP) and has fewer data requirements (Horritt and Bates, 2002; Teng et al., 2017). HEC-RAS can also be adequately calibrated on hydrometric data and has proved to accurately predict flood extent when water surfaces are superimposed onto accurate high-resolution digital elevation models (DEM). For these reasons, HEC-RAS has been
applied worldwide for flood simulation purposes, especially in locations highly prone to suffer flash floods, including Vietnam (Nguyen et al., 2015), Malaysia (Alaghmand et al., 2012), Pakistan (Khattak et al., 2016), Turkey (Curebal et al., 2016), Greece (Papaioannou et al., 2016), Spain (Ruiz-Villanueva et al., 2013) and the United States (Knebl et al., 2005).

A LIDAR (Laser Imaging Detection and Ranging) derived Triangular Irregular Network (TIN) has been generated to extract detailed floodplain topography, so as to provide HEC-RAS with reliable topographical information to accurately delineate the

cross-sections needed to set up the hydraulic model. To do so, eight 2x2 km sheets of LIDAR point clouds from the Spanish National Geographic Institute (http://centrodedescargas.cnig.es/CentroDescargas/locale?request_locale=en) have been obtained, covering Sant Llorenç town and its surroundings upstream, with an average point spacing of 0.82 m. A 1 m resolution DEM based on the returns of the laser beams generated by bare ground, buildings and roads was then created. However, due to the noise contained in some LIDAR point clouds, the delineation of cross-sections and river bed elevations based on LIDAR-derived DEMs may favour the introduction of artefacts, which could cause the underperformance of the hydraulic model. To avoid this, a TIN model has been generated, by using the detailed contours delineated at intervals of 1 m on the LIDAR-derived DEM, which serves to better define floodplain topography for flood simulation purposes, since TIN-derived cross-sections present smoother curves (Costabile and Macchione, 2015). The HEC-RAS model has been set up using 40 cross-sections distributed along the 4303 m of reach length, to provide the model with information on channel width and bed elevation. The number of cross-sections was limited by the meandering shape of the Ses Planes river at Sant Llorenç and the extraordinary extent of flooding in the city centre (cross-sections wider than 300 m were required to cover the affected areas at some points), since cross-sections could not intersect each other. For bridge modelling, two additional cross-sections were defined, located immediately upstream and downstream of each of the seven bridges crossing Ses Planes stream channel in Sant Llorenç town. Upstream boundary conditions for the simulation were imposed by the FEST-modelled discharge hydrograph at the first cross-section, and downstream boundary conditions were imposed by the frictional slope calculated empirically from TIN-derived river bed slope between cross-sections. Manning's $n$ roughness coefficient values (Table 2) were based on CORINE land cover data from 2018 and adapted following Papaioannou et al. (2018). Detailed topographic information on floodplain was set up for the analysis using HEC-GeoRAS extension for ArcGIS®, and the topographic and geometric information was depurated using HEC-RAS software in an iterative fashion. This process included the deletion of topological errors in river banks and hydraulic incongruities at bridges, and the filtering of cross-section delineating points.

For dynamic reconstruction of the flash flood (i.e., its timing and the spatial extent of flood-affected areas), HEC-RAS unsteady flow analysis was applied. The plan established for this flash flood simulation started at 1700 h and ended at 2400 h LT on October 9, ensuring that the entire event, as pinpointed by the FEST-modelled discharge hydrograph, was encompassed within the analysis. Since a strong agreement was found between the performance of FEST and KLEM, increasing the confidence in the numerical results in terms of peak discharge and timing, only the former simulated discharge has been used to force HEC-RAS. The computation time step was set up at 10 minutes (same temporal resolution as the discharge hydrograph) and the output time interval was set at 1 minute, although only flooding maps (1 m resolution) every 10 minutes are shown. Validation of the results was made by comparing the simulated flooded area with the observed event by Copernicus Emergency Management System (Sentinel-1 imagery), and the modelled water depth was compared with 32 flooding marks measured in-situ a few days after the event.

**3.5 Evaluation of flash flood magnitude and geomorphological impacts**

In order to contextualize the exceptionality of this flash flood and for comparison purposes, a power metric that summarizes the magnitude of the event was used. One option for quantifying the power of moving water that drives sediment transport and geomorphic change is the unit stream power (Wm-2). It is computed as


$$\omega = \frac{\gamma Q S}{w} \qquad\qquad (7)$$

where γ is specific weight of water (Nm-3), Q discharge (m3s-1), S is friction slope (m/m, here equal to the channel slope). These three components are the total stream power (Ω) that is normalised by peak flow width (w; Bagnold, 1966).

For assessment of the geomorphological effects of the event, several remote sensing techniques were used. First, two false infrared RGB (red, green and blue) composites of Planet® high resolution (3 m pixel size) spectral imagery were visually compared, acquired hours before and the day after the event. In these image composites, the band corresponding to the short wave infrared (SWIR) is set to the red channel. Thus, the increasing signal of the red channel (pink) in the image after the flash flood denotes a rise in the responsive signal of pixels corresponding to bare soil, rocks and sediment (Lillesand et al., 2015),

suggesting the occurrence of strong runoff processes with associated mudflow and sediment erosion and transport. Consequently, a huge amount of debris was deposited in several new gravel and coarse sediment bars along the Ses Planes torrent floodplain. An estimation of the sediment deposited was made using a very high resolution DEM (6 cm pixel size) and orthophotograph (3 cm pixel size) calculated using Structure from Motion (SfM) techniques (Schumann et al., 2019). Both corresponded to an external Unmanned Aerial Vehicle (UAV) photogrammetric flight (Garau Ingenieros®) from November

305 2018.

For quantification of sediment extent, a two-step approach was used, based on the abovementioned very high resolution imagery and elevation data. First, image spectral analysis was used to differentiate among different cover types, and an interactive supervised classification was performed, manually defining the training samples on the RGB bands of the orthophotograph (Camenen et al., 2013). Spectral-based classification performed well when classifying vegetation types and

water cover, although sediment spectral responses in the RGB bands were not enough distinguishable to correctly discriminate between sediment types (not shown). To overcome this, the procedure described in Carbonneau et al. (2005) was followed, using texture as an indicator of sediment type and its spatial patterns. Orthophotograph pixels were evaluated in a Grey Level Co-occurrence Matrix (GLCM), used to quantify how many pixels of similar grey levels are neighbours. This windowing (kernel) approach generates a textural image which retains local image texture properties using contrast calculation and allows

classification of pixels into different types (land covers) based on very high resolution (3 cm in this case) images. This textural image was used to accurately classify land cover into six categories (trees, shrubs, herbaceous vegetation, water, fine sediment and coarse sediment) by means of an interactive supervised classification (manually defining the training samples on the textural image). More details about GLCM calculation can be found in Carbonneau et al. (2005).

Once the extent of the sediment deposited during the event was obtained, the height of the event-created gravel bars and deposits were estimated by subtracting the height of each pixel in the LIDAR-derived DEM (data acquired before the event) from those in the SfM-derived DEM (data acquired less than one month after the event; Fig. 8c and 9). For volume calculation, the base area (each 6 cm² pixel) was multiplied by the height of each sediment bar pixel calculated before. In this way, each (2-D) pixel has an associated (3-D) volume which corresponds to the column above it and is defined by pixel bar height (a rectangular prism in which the base is the square pixel and the side is the height value). To compute an estimation of bar sediment weight, the volume raster layer per rock density (2700 kg/m³ for limestone) and porosity of the deposit (15%) were multiplied.

## 4. Results

### 4.1 Convective precipitation predictability

A first set of experiments consisted of nesting the WRF domains in the deterministic high resolution (~9 km horizontal) ECMWF atmospheric model using the 0000 h and 1200 h UTC 8 October (0200 h and 1400 h LT, Local Time), and 0000 h UTC 9 October operational cycles of the EPS. Although all three simulations show precipitation accumulations exceeding 100 mm somewhere in the region, none of them produced rainfall above 50 mm for the afternoon and evening of October 9. This reveals the presence of forecast errors from sources other than the resolution limit (Fig. 2c and 2d).

The 12h-accumulated precipitation valid at 0000 h UTC 10 October for all members showed a high diversity of possible scenarios but none of them showed enough ability to have triggered dependable warnings over the affected area. Indeed, the field of probability of 12 h-accumulated precipitation above 100 mm focused attention over the western side of the domain, with only marginal probabilities over the eastern sector of Mallorca and negligible odds at the Ses Planes catchment (Fig. 2c).

### 4.2 Precipitation estimates

Figure 3a shows the estimated cumulative precipitation during the day of the flood event in northeast Mallorca, where the Ses Planes catchment is located. Radar-based estimates of precipitation have been compared to rain gauge observations, showing an excellent agreement: squared correlation coefficient is slightly above 0.98 and radar-derived precipitation features a small positive bias of 2.5 mm when averaged over the 21 rain gauges (Fig. 3b). The spatial signature of the moving train of convective cells precipitating again and again over the same region is clearly visible from the distribution of cumulative radar-derived precipitation (Fig. 3a and 4). The train of convective cells followed a south-to-north direction, affecting Ses Planes basin from 1300 h to 2000 h UTC (1500 h to 2200 h LT) on October 9, 2018. As a result, an area larger than 100 km2 (Fig. 3c) was affected by an intense rain of 200 mm, with increasing values towards the centre of such area, reaching peaks of 350-400 mm in the upper parts of the Ses Planes catchment, which features fan-shaped topography. This part of the catchment lies within the isohyetal line of 300 mm and, with an extension about 10 km2 functioned as a funnel, gathering the precipitation and generating

intense runoff processes that rapidly accumulated flow into the well-developed hydrological network. The most intense
precipitation period over the catchment lasted four hours (from 1600 h to 2000 h LT approx., Fig. 4), which gives an average intensity of around 60 mm per hour. However, peak rainfall exceeded 100 mm h$_{-1}$ in the moments previous to the flood event, with 140 mm falling between 1840 h and 1940 h LT that added up to the 130 mm fallen in the previous hours (Fig. 3d and 4). Note that LT (Local Time) corresponds to UTC plus 2 h.

## 4.3 Coupling hydrological and hydraulic modelling of the event

After calibration with in-situ measurements, FEST and KLEM have simulated peak discharges around 305 m$_3$s$_{-1}$ at the entrance of Sant Llorenç town (Table 3). According to the hydrological simulations, peak discharge occurred at 1920 h LT in the reach 2 km upstream of Sant Llorenç and between 1930 h and 1940 h LT at the entrance of the town. A high inter-model consistency has been found between the values of the different infiltration and dynamical parameters (Table 1). Overland flow velocities ranged between 0.35 and 0.40 m s$_{-1}$, while channel flow speeds varied from 3.3 to 4.1 m s$_{-1}$, highlighting the sharp hydrological response of the Ses Planes basin after torrential precipitation.

Almost 400 mm of rainfall accumulated in barely six hours in the fan-shaped headwaters of the watershed (Figs. 3 and 4). In spite of the large amount of water lost via infiltration and deep percolation (Tables 1 and 3), a significant amount of overland flow was rapidly routed into the river network and streamflow, which was inconsequential minutes before, rapidly increased. The hydraulic simulation shows how water started flowing through Sant Llorenç town at 1900 h LT and, in barely 10-20 minutes, overflooded the artificial channel at several points within the sector between bridge #2 and bridge #3, reaching depth values of about 3 m (Fig. 5). The football field, located in the meander between bridges # 2 and # 3, was completely flooded in less than half an hour. It only took between 10 and 20 minutes for the flood modelled at the entrance of the town to produce the maximum extent of affected area in the city centre: water covered the entire longitudinal path of the channel across Sant Llorenç, with the most affected areas in the vicinities of the bridges and river corners. The major peak wave entered the northern sector of the town centre around 1940 h LT, completely overflowing bridge #3 and adjacent areas and began to flood the Town Hall square (the very town centre) with around 0.50 m of water. As the peak flow wave moved southwards, the southern sector of the town centre between bridge #3 and bridge #4 was severely affected. Water depth reached almost 1.5 m in the Town Hall square and the surrounding blocks, which are located 150-200 m away from the main river channel. After 2000 h LT the floodwaters receded from most of the town, except for an area located between bridge #3 and bridge #4, which remained flooded until 2030 h LT, due to the backwater effect of bridge clogging and the wide open topography of the surrounding non-built area. According to this numerical simulation, Sant Llorenç was only flooded during barely 1 hour, which gives an idea of the sudden character of this flash flood event.

## 4.4 Flood simulation and flood area mapping

The simulated water depths from the modelled discharge and detailed flood extent have been compared against the observed
flooded area by Copernicus Emergency Management System (via Sentinel 1 radar imagery, Fig. 6). Furthermore, the modelled
water depths have been compared against the observed flooding marks from on-site field measurements (Fig. 6d). The last
comparison exhibits a high correlation coefficient ($R_2 = 0.91$), indicating a good performance and reliability of the hydrological
and hydraulic simulations. Figure 6a shows how the simulated flooded area mostly corresponds to that estimated via satellite,
except for one section located about 800 m upstream of the town entrance. This discrepancy can be explained by the flood
breaking down a stone wall and inundating the lands beyond, and this was not accurately reproduced by the hydraulic
simulation. Figure 6a also shows the area corresponding to theoretical floods for return periods of 10, 100 and 500 years, given
by the Spanish Ministry of Agriculture, Food and Environment for this particular area (MAPAMA, 2010). The estimated
flooded area by both the hydraulic simulation and the observed event by Copernicus Agency goes far beyond the floodplain
for a 500-yr return period, indicating the exceptional nature of this event. The detailed maps (Fig. 6b and 6c) show that the
inundated areas near the channel, which included residential buildings, reached water depths close to 2 m. This figure entirely
agrees with in-situ measurements taken inside the houses of various residents. Of note is the large extension of the flooded
area inside the town (up to 100 m from side to side in some areas), compared to that which occurred upstream in agricultural
lands (around 60 m width), as section plots in Figure 7 depict. The most affected areas were those located between bridges and
channel corners, due to the clogging effect they exerted on water flow. The presence of several bridges also affected the
dynamical behaviour of the flash flood, as is very well observed in Figure 7, where water velocity from the simulation is
depicted. Water flowed down the natural riverbed up to where bridge #2 is located, and where the concrete artificial channel
starts (it is worth noting that the artificial channel capacity is 161.5 $m_3 s_{-1}$, whereas a flood peak discharge around 305 $m_3 s_{-1}$
was estimated). That location also coincides with a 105° turn in the riverbed as it is canalised; therefore, the water was forced
to turn and pass through a narrower section, where the roughness of the riverbed greatly decreases. The consequence of this
was a vast increase in channel flow velocity from roughly 3-4 m $s_{-1}$ before the corner to 7 m $s_{-1}$ after the corner. From there,
the topography opens up to a plain zone where water floods and velocity decreases. The other major increase in velocity
occurred right after the second corner upstream of bridge #3 (Fig. 7). This bridge occupies more than half of the flooded
section, thus blocking the water and increasing its velocity upstream due to flux accumulation and generation of intense
backwater effects caused by bridge clogging. The same happened upstream of the other bridges (minor orange-red spots in
Fig. 7) but of a smaller magnitude.

## 4.5 Flash flood magnitude and geomorphological impacts

The estimated peak discharges have a corresponding unit peak discharge slightly above 13 $m_3 s_{-1} km_{-2}$, and a lag time of 0.5-
0.8 hours (Table 3). These estimations fits well with the relationship between lag time and catchment area proposed by Marchi
et al. (2010) for areas $< 350$ $km_2$:

$$T_L = 0.08A^{0.55} \qquad\qquad (6)$$

with a result of 0.45 hours (27 minutes). Note that lag time has been computed as the time difference between the net rainfall and runoff centroids. The unit stream power obtained for the natural flooded section was 1,110 Wm$_{-2}$. The value can be

considered in the upper range for this type of small catchment with steep slopes. The magnitude of the flash flood impacted the channel, changing its morphology. High resolution satellite images before and after the event show the geomorphological impacts over a 4.4 km channel upstream of the town (Fig. 8). A volume of 14,000 m$_3$ of sediment (5,671 metric tonnes) was deposited as new bars, especially on the meanders. The sediment was composed of gravel and fine material, with sporadic blocks on top of some deposits. Some areas accumulated maximum sediment depths of 1 m. The flash flood eroded sediment

from the agricultural fields on the headwaters, especially on those located over softer lithology (Cretaceous marls and limestones) in the western area. The flood also eroded shrubs and numerous trees along the channel. Figure 9 shows what occurred in the largest meander on the studied reach. The concave area of the meander facilitated deposition of a large amount of transported sediment, although woody debris enhanced the aggradational process. The original main channel moved 30 m northwards after the flash flood, with a deposition of more than 1,000 tonnes of sediment in a new fluvial stream bar (Fig. 9c).

Comparing Figures 9a and 9b, it is possible to identify how several trees disappeared after the flood. The new bar created by the flash flood is up to 200 m length and around 65 m width, reaching 1 m depth at some points. This new channel created after the flash flood is a clear example of the geomorphological effects of this large magnitude event.

## 5. Discussion

### 5.1 Predictability and hydro-meteorological modelling issues

This singular case exemplifies the complex challenges faced by scientists, hydro-meteorological forecasters, civil protection managers and policy makers in predicting Mediterranean flash floods. Without a doubt, the main challenges of prediction in this instance were the small spatial and temporal scales of multiple elements, which contributed specifically to the tragic urban flash flood of October 9, 2018 at Sant Llorenç. Determinant precipitation structures at the sub-kilometric scales in the convective systems affecting the eastern Balearic Islands were identified in previous sections, severely impacting the

hydrological catchment. Among conventional precipitation prediction methods, statistical techniques using historical records, such as analogs (Hamill et al., 2006), pattern-based (Nuissier et al., 2011) or statistical downscaling of large scale forecasts (Wilks, 2010) are of very limited value for the warning protocols over useful forecast ranges. The inability of the numerical weather prediction models to sufficiently forecast the location, intensity and timing of the precipitating systems that triggered the tragic flood on October 9, 2018 in the village of Sant Llorenç, outlines important challenges and research questions for the

hydro-meteorological communities. Likewise, when model errors are also considered by activating stochastic perturbed

parameters in the boundary layer and land surface parameterizations in WRF (Jankov et al., 2017), location and intensity of predicted torrential rain cells are not significantly improved (not shown). However, regarding precipitation amounts and the usable limits of these 12 h to 24 h forecasts for early warnings and civil protection awareness, these exploratory ensemble prediction systems show the possibility of maximum 12 h accumulations exceeding 250 mm at distances below 30 km from the St. Llorenç catchment for this particular event. Given the aforementioned characteristics of the catchment and the fairly typical location errors of determinant convective cells obtained for this case, these state-of-the-art sub-kilometric ensemble predictions may well illustrate a serious predictability limit of precipitation structures usable in hydrological forecast systems over 12 h to 24 h lead times. In this context, an ideal hydro-meteorological forecasting system suited for sub-urban scales should combine an accurate sub-kilometric representation of the precipitating systems, decametric resolution in the hydrological modelling and metre or sub-metre precision in the hydraulic component of the forecasting chain. These ambitious spatial requirements are linked to, and thus also determine, the temporal scales accounted for in each modelling phase.

## 5.2 Radar-derived rainfall estimates

Despite predictive models failed to foresee and reproduce the huge amount of precipitation recorded in the north-eastern part of Mallorca during October 9, 2018, we were able to accurately estimate it using radar information. However, the high spatial variability and uncertainty associated to convective storms introduces limitations when deriving precipitation estimates with radar at long distances (Burcea et al., 2019). Fortunately, the Doppler weather radar is located 60 km from the catchment, allowing us to obtain very accurate precipitation estimates. This is important, since the region features an unevenly distributed meteorological network and, more specifically, for the studied catchment which, despite several flash floods occurred over the last decades, still lacks automatic weather and gauging stations for rainfall and streamflow data recording. A comparison of the areas over which a given amount of accumulated precipitation was exceeded, according to Sant Llorenç' surrounding rain gauges and radar measurement devices, reveals that the sampling characteristics of the former network were not small enough for a correct delineation of the heavy rainfall areas (Fig. 3c). The highly local spatial scale at which heavy rainfall developed prevented its suitable observation by the relatively dense but irregularly distributed pluviometric network (Marchi et al., 2010). In this context, accurate reproducibility of precipitation amounts generated by intense convective rainfall episodes in ungauged catchments using radar data, represents a significant contribution for regions with sparse or uneven distribution of meteorological stations.

The total precipitation volume during the event, with an areal basin average close to 317 mm, was three times larger than the monthly climatological precipitation for October, recorded in the nearby town of Artà (100 mm yr$_{-1}$), and just half of the annual climatological precipitation (696 mm yr$_{-1}$; climatic average 1974-2014). Furthermore, it clearly surpassed the daily precipitation value (210 mm day$_{-1}$) for a return period of 500 years, estimated by the local government of Mallorca island for that particular area (DGRRHH, 2001), indicating the exceptional nature of the event.

The question that remains is why such amounts of precipitation fell over that catchment and not on nearby areas. Two hypotheses may explain this phenomenon: on one hand, the existence of a stationary convergence line that favoured convection on that particular south-north fringe; on the other hand, the anchoring effect of the orography, slowing down the movement of the convective cells train. However, the mountains of the catchment are not high enough to produce such an anchoring effect, thus the role of orography was limited to just enhance the convection uplift as the convective cells moved northwards.

The very accurate radar-derived precipitation estimates obtained were the main input of the hydrological models, and this represents a vital step in the reconstruction of the event. The robustness of the method used and the accuracy of the estimations increased the reliability of the subsequent hydrological and hydraulic simulations. Nevertheless, it is necessary to reduce processing time and facilitate access to near real-time information broadcasting for meeting early warning purposes. This is operatively complex and implies investment, both in working force and in funding.

## 5.3 Hydrological modelling

Comparison in the performance of the simulated hydrographs by two distinct hydrological models has served to better quantify the associated uncertainties when simulating peak discharges and flow velocities for this extreme event. The strong inter-model output consistence highlights the basic equivalence of the numerical simulations for the 9 October 2018 extreme flash-flood when considering different degrees of model complexity.

The simulated peak flows are close to the expected discharge for a 500-year return period and depict specific peak discharges slightly above 13 $m_3s_{-1}km_{-2}$ (Table 3). The extremely sharp rising limbs denote a very fast time to peak (50 minutes in the case of FEST), which may be explained by several factors such as: sparse and low density vegetation, steep slopes, thin soils and watershed topography. These physical characteristics, together with the overwhelming rainfall intensity resulted in fast Hortonian flows that favoured the rapid generation of streamflow with lag times shorter than one hour (Table 3), increasing the hazardous component of the forthcoming flood.

Simulated unit peak specific discharges for the October 9 2018 (~ 13 $m_3s_{-1}km_{-2}$) event were similar to those recorded in other catastrophic flash floods that have occurred in the Mediterranean basin. Delrieu et al. (2005) indicated specific peak discharges between 10 and 30 $m_3s_{-1}km_{-2}$ in several watersheds of the Mediterranean Gard region in France (the so-called *Gard event*) during a catastrophic flash flood event that occurred on September 8 and 9, 2002. This flood was at a larger scale (in terms of rainfall amount and the extent of the affected areas) than the flood at Sant Llorenç. Furthermore, the estimated specific peak discharge for the Sant Llorenç flash flood is one order of magnitude larger than those recorded by Tarolli et al. (2012) in selected Mediterranean small- and medium-sized basins of Italy, France and Spain.

The modelled hydrographs for the October 9 2018 event exhibit very steep rising limbs that fall in less than two hours. These features are similar to other observed hydrographs of significant flash floods over Spain, such as the region of Valencia (Camarasa-Belmonte and Segura-Beltrán, 2001), the Gredos Range in central Spain (Ruiz Villanueva et al., 2014), or the lower

reaches of the Guadiana basin (Ortega and Garzón-Heydt, 2009). However, in most cases, both absolute and specific discharges generated during these flood events were much lower than in the case of Sant Llorenç.

## 5.4 Hydraulic simulation performance

The hydraulic model has simulated a flood extent very similar to the one delineated by the Copernicus Emergency Management System. The main differences have been an underestimation of the modelled flood extent in the reach immediately before the entrance of Sant Llorenç, due to a break in a wall, and the greater extent of the modelled flood in the north-eastern quadrant of the town centre. The latter may not be an overestimation as several Sant Llorenç locals confirmed that those areas were flooded during the event. Moreover, 32 in-situ after-event flooding marks have been successfully compared with the simulated

depths. Therefore, the hydraulic simulation forced with the FEST model has been very accurate in terms of flood spatial extent delimitation (Papaioannou et al., 2016), serving as an additional verification check for the hydrological experiments.

The larger extent of the flooded area at Sant Llorenç compared to the reach upstream the town can be explained by several factors: the aforementioned effect of bridges and channel corners which blocked the water and the transported sediment and debris; the topography, as the town is located in the flattest (therefore susceptible to be flooded) area; and finally, the effect of

concrete and pavement, which prevented infiltration and increased the kinematic energy of the water body and consequently the destructive power of the flash flood.

The simulated flow velocities introduce more uncertainties since the results cannot be verified by comparison with direct measurements. However, flow velocities simulated with both the hydraulic and the hydrological models, are consistent with those found by Ruiz Villanueva et al. (2014), who obtained flow velocities above 3 m $s^{-1}$ for a peak flow discharge of 120 $m^3 s^{-1}$

(less than half of the estimated peak discharge at the entrance of Sant Llorenç) for a catchment of 15.5 km² in central Spain. They also observed how flow velocity highly increased at bridge locations and generated intense backwater processes. Additionally, Yalcin (2018) simulated maximum velocities around 4 m $s^{-1}$ for an expected peak flow of 25 $m^3 s^{-1}$ in a 11 km² catchment in central Turkey. In this context, the use of the 1-D approach for the analysis of bridge effects may not properly simulate the interactions between bridges and flow, and backwater effects may be underestimated or neglected (Costabile and

Macchione, 2015). This seems not to be the case for the simulation of the Sant Llorenç event since flow velocity greatly increased at all bridges. In addition, the sections between bridges #3 and #4 were flooded for a longer time, denoting that the model successfully modelled the occurrence of intense backwater processes at these locations.

## 5.5 Flash-flood magnitude and geomorphological impacts

Estimated peak discharges, unit peak discharges and lag times give an idea of the extraordinary and sudden character of the

flash flood that occurred in the Ses Planes ephemeral stream. The unit stream power of 1,110 $Wm^{-2}$ is related to bank erosion and deposition and to the risk of substantial channel widening or major geomorphic change (Yochum et al., 2017). This value is similar to other flash flood events which caused a major geomorphic response in an alluvial channel, reported for small

streams in the Mediterranean climatic region (Marchi et al., 2016). Although the analysed stream features steep slopes and has an area similar to other studied catchments, several studies reported one order of magnitude more on unit stream power for the latter events (i.e., Baker and Costa, 1987; Batalla et al., 1999).

After the flood, a significant trend in channel widening was observed, similar to other stream floods (Righini et al., 2017), with a large accumulation of gravel and fine material in new fluvial bars. The accumulation of woody debris decreased transport capacity, and flow obstruction by vegetation created a positive feedback for instream aggradation (Merritt and Wohl, 2003; Ruiz-Villanueva et al., 2013). Along the studied reach, several trees were eroded, transported and deposited downstream, arriving at the first bridge at the entrance of the village, facilitating channel widening and erosion of the initial artificial channel, and also increasing the hazardousness of the flood peak wave, which featured a high destructive power (unit stream power > 1,000 Wm$_{-2}$).

Even so, it is important to clarify that the 9 October 2018 flash flood was not a hyperconcentrated debris flow. The transported sediment had a negligible impact on the peak discharges or timings. This fact is confirmed by the subsequent hydrological experiments. In-situ post-event field works have estimated a total deposited sediment volume of 1.4·10$_{-2}$ Hm$_3$, while the hydrological simulations have yielded a total runoff volume ranging between 1.0 to 1.9 Hm$_3$ (Table 3). Therefore, sediment concentration was at most 1.4% of the total water volume, well below the concentration threshold of 6% from which water flood behaviour begins to be affected by sediment (Pierson, 2005).

## 6. Conclusion and outlook

In studying the October 9, 2018 flash flood at Sant Llorenç, it can be concluded that almost all the possible hazardous factors converged to generate this extraordinary event and its tragic consequences. Contributing factors were the inability of predictive models to foresee the convective heavy rainfall event, the synergistic effect of the storm cells' trajectory, and topography that enhanced convection in the headwaters of the catchment (insight on this matter could be provided by fine-scale numerical simulations in further research). Additional contributing factors were the very small drainage area owing a very short hydrological response time favoured by several ingredients, such as: steep slopes, funnel-shaped watershed, low vegetation density and thin soils. These basin features favoured the acceleration of the runoff processes. Further contributing factors were the location of the town in the floodplain of the Ses Planes torrent, the artificial channelization of the stream at the entrance of the town, the presence of several bridges that acted as stoppers for flow and debris and generated intense backwater effects, a delayed red alert warning, and low awareness and conscientiousness of people. Figure 7 perfectly illustrates this last point as people crossed bridge #3 seconds before the water flooded the road above it. As the IPCC (2012) reported, the increase of natural risks worldwide is more related to increased vulnerability (settlements and goods located in the river floodplains), socio-economic impacts (land use changes) and perception (low awareness and preparedness), rather than to increased hazardousness. This idea is corroborated with the analysis performed in this study. This extraordinary convective rainfall event (> 300 mm day$_{-1}$ estimated versus 210 mm day$_{-1}$ expected for a 500-year return period), and the magnitude (305 m$_3$s$_{-1}$ modelled

versus 348 m3s-1 expected for a 500-year return period) and impacts of the flood were aggravated by human-dependent factors. This relationship between intense rainfall events and increased hazardousness due to anthropogenic factors has become a dominant pattern in the Mediterranean region (Llasat et al., 2014) and it represents a major setback for effective flash flood predictability and warning.

A question remains with respect to both the maximum lead time and precision in the characterization of this flash flood, linked

to the location, intensity and mode of the precipitating systems responsible for this tragic event. This case study has revealed the limited accuracy of the numerical weather prediction models when locating torrential precipitations 12 h to 24 h ahead. These predictability limits point out to the need that warning systems and civil protection activation protocols can cope with forecast errors in the range of 30 to 50 km. As important as it is to focus research efforts on characterizing and reducing these errors, it is also important to consider these errors in future warning frameworks to come.

*Author contributions*. CG, JLL, EMT, VH, AA, CR, and RR designed the research. AA, JLL, VH, AMF, AH, CR and RR performed the formal analysis. CG and EMT completed the field work campaign. JLL drafted the paper. JLL, AA, and AMF elaborated the figures. All authors contributed to the writing, editing and revision of the paper.

*Competing interests*. The authors declare that they have no conflict of interest.

*Acknowledgements*. This work has been sponsored by CGL2017-82868-R [Severe weather phenomena in coastal regions: Predictability challenges and climatic analysis (COASTEPS)], which is partially supported by Fondo Europeo de Desarrollo Regional (FEDER) and Secretaría de Estado de Investigación, Desarrollo e Innovación. Author A. Hermoso received funding from the Spanish Ministerio de Educación, Cultura y Deporte (FPU16/05133). The authors thankfully acknowledge the computer resources at MareNostrum 4 and the technical support provided by Barcelona Supercomputing center (RES-AECT-
595 2018-3-0009 and RES- AECT-2019-1-0008). The authors thankfully acknowledge the spectral high resolution images ceded by Planet®. The very high resolution orthophotograph and digital elevation model were ceded by Garau Ingenieros® working in a reconstruction project promoted by Direcció General de Recursos Hídrics del Govern de les Illes Balears. The authors want to thank the interviewed Sant Llorenç locals for their testimonies, pictures and videos taken during the event. The authors also want to thank the two anonymous reviewers for their comments aimed to increase the quality of the paper.

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

| Catchment | Area (km²) | Total rainfall (mm) | CN (II) | $S_0$ (mm) | λ | $V_h$ (ms⁻¹) | $V_c$ (ms⁻¹) |
|---|---|---|---|---|---|---|---|
| Ses Planes | 23.2 | 316.7 | 63.4 (10.0) | 431.8-457.2 | 0.35-0.40 | 0.36-0.40 | 3.3-4.1 |

**Table 1** Range of the calibrated parameters according to FEST and KLEM hydrological models. Total rainfall amount and curve numbers are expressed as area-averaged values. CN standard deviation is shown between brackets. Note that initial CNs correspond to normal antecedent conditions. $V_c$ corresponds to the maximum channel velocities for FEST.

| Land cover | Manning's n |
|---|---|
| Broad-leaved forest | 0.1 |
| Complex cultivation patterns | 0.04 |
| Coniferous forest | 0.1 |
| Discontinuous urban fabric | 0.013 |
| Fruit trees | 0.08 |
| Land principally occupied by agriculture | 0.05 |
| Mixed forest | 0.1 |
| Non-irrigated arable land | 0.05 |
| Sclerophyllous vegetation | 0.05 |
| Transitional Woodland-shrub | 0.06 |

**Table 2** Manning's n roughness coefficient values used for flood modelling (adapted for CORINE land cover categories; Papaioannous et al., 2018).

| Catchment | Area (km²) | Total rainfall (mm) | Peak discharge (m³s⁻¹) | Timing (UTC) | Runoff volume (Hm³) | Unit peak discharge (m³s⁻¹) | Runoff ratio (-) | Lag time (h) |
|---|---|---|---|---|---|---|---|---|
| Ses Planes | 23.2 | 316.7 | 303.4-306.9 | 17:30-17:40 | 1.0-1.9 | 13.1-13.2 | 0.13-0.26 | 0.5-0.8 |

**Table 3** Range of the main hydrometeorological features estimated from the hydrological simulations for the 9 October 2018 flash-flood episode at the entrance of Sant Llorenç town. Note that lag time has been computed as the time difference between the net rainfall and runoff centroids.



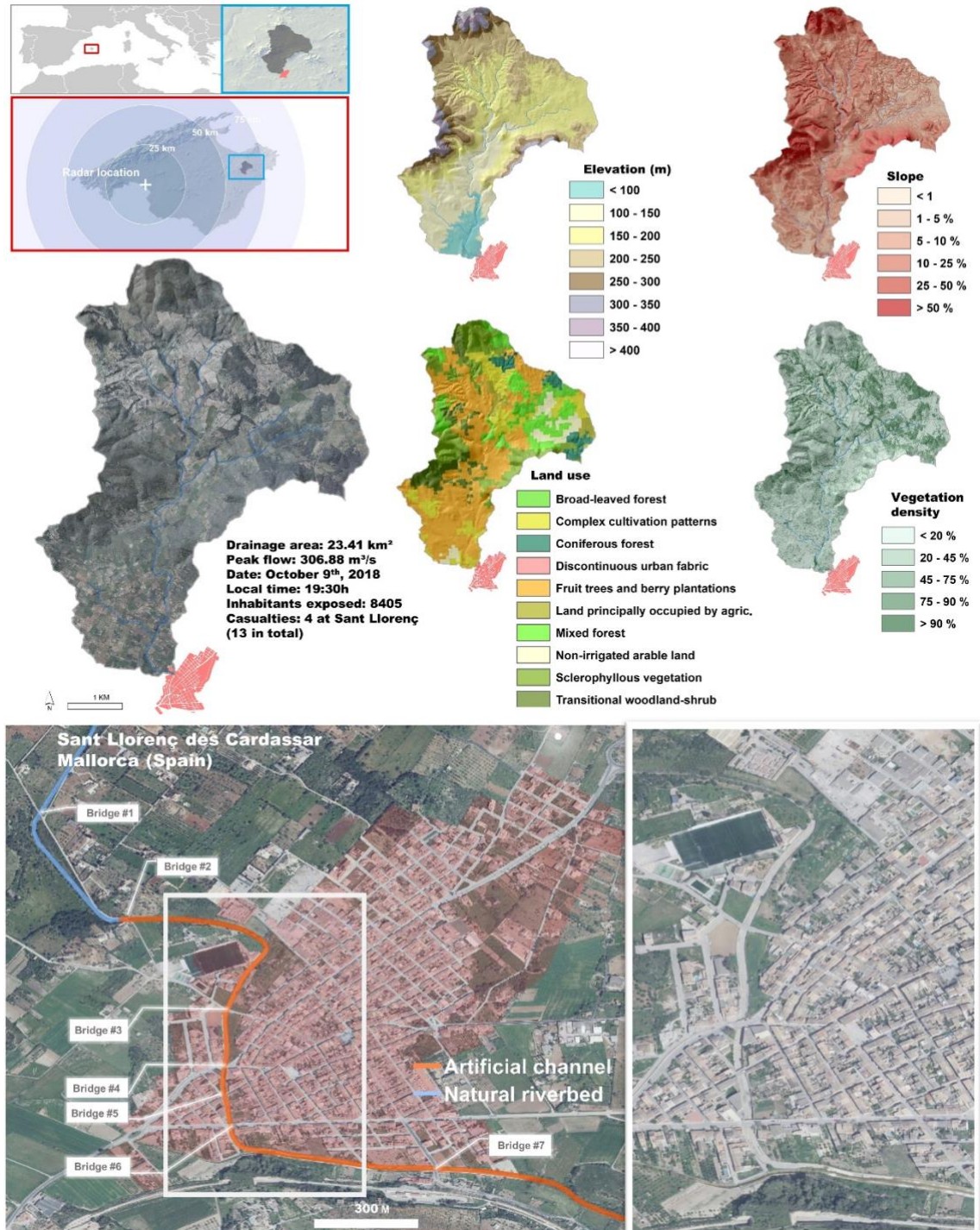

**Figure 1**. Study area location and overview. Upper panels: characteristics of the contributing watershed at Sant Llorenç (elevation, slope, land cover and LIDAR-derived vegetation density). Lower panels: detailed views of the urban interphase of the torrent de ses Planes at Sant Llorenç des Cardassar.


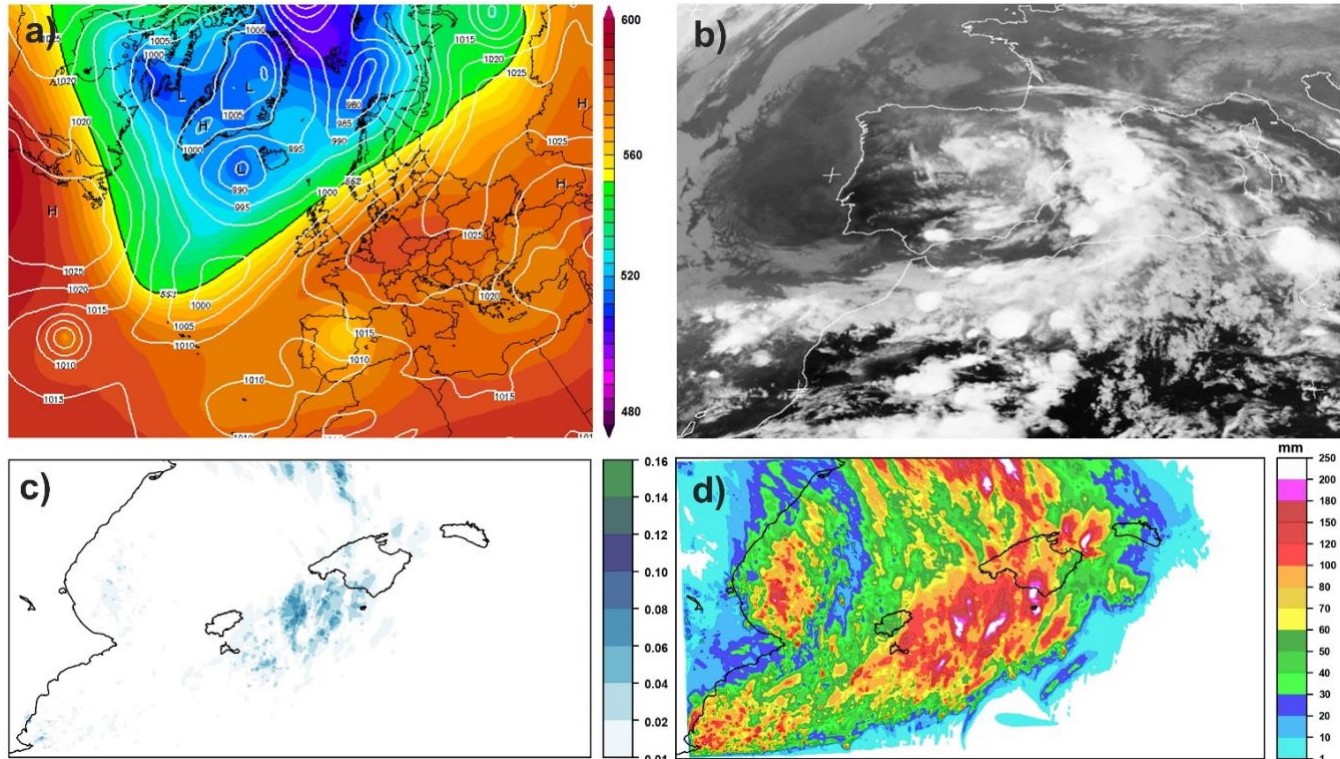

**Figure 2.** (a) Synoptic situation on October 9, 2018 at 1200 h UTC (1400 h local time) from CFS reanalysis. Colors show geopotential height at 500 hPa (in gpdam according to scale). White contours are isobars (hPa) at sea level (source: https://www.wetterzentrale.de; CC BY-NC). (b) IR MSG image corresponding to October 9, 2018 at 1700 h UTC. (c) Probability of 12h-accumulated precipitation exceeding 100 mm valid at 0000 h UTC October 10, 2018 from the larger WRF domain initialized at 0000 h UTC October 9. (d) Maximum 12h-accumulated precipitation across ensemble members valid at 0000 h UTC October 10, 2018 from the larger WRF domain initialized at 0000 h UTC October 9.

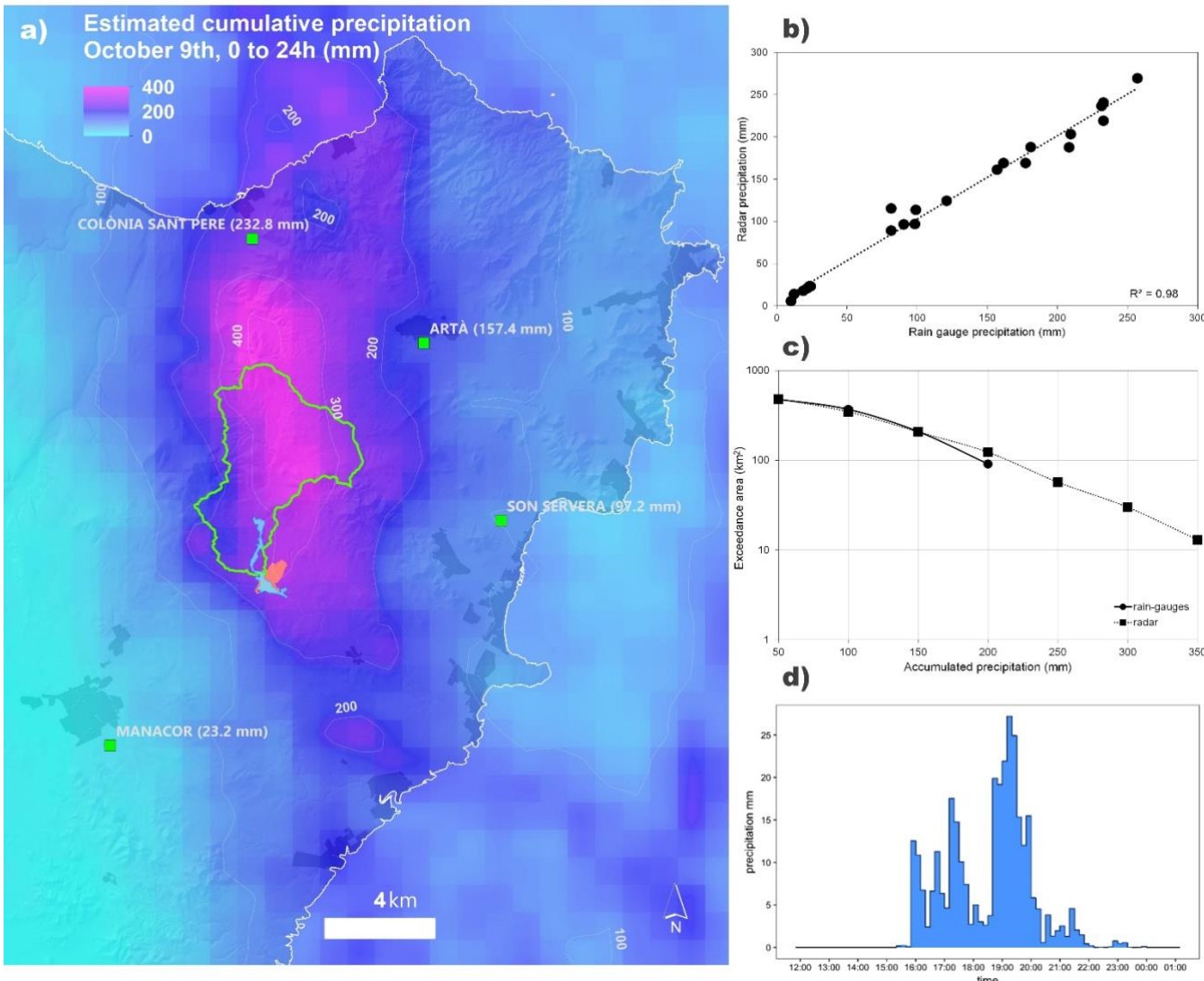


**Figure 3**. (a) Spatial distribution of the 24-h estimated cumulative precipitation for the October 9, 2018 flash flood. Automatic rain gauges used for bias correction are shown as squares. Circles depict the daily pluviometric stations. (b) Scatterplot of the 24-h accumulated precipitation derived from radar estimates and pluviometric stations. (c) Comparison of the curves of excedance areas for different rainfall thresholds derived from the 24-h accumulated pluviometric and radar-based rainfall

amounts. All pluviometers shown in Fig. 3a have been used for the computation (21 stations). (d) Radar-driven precipitation for the October 9, 2018 episode at Ses Planes catchment.

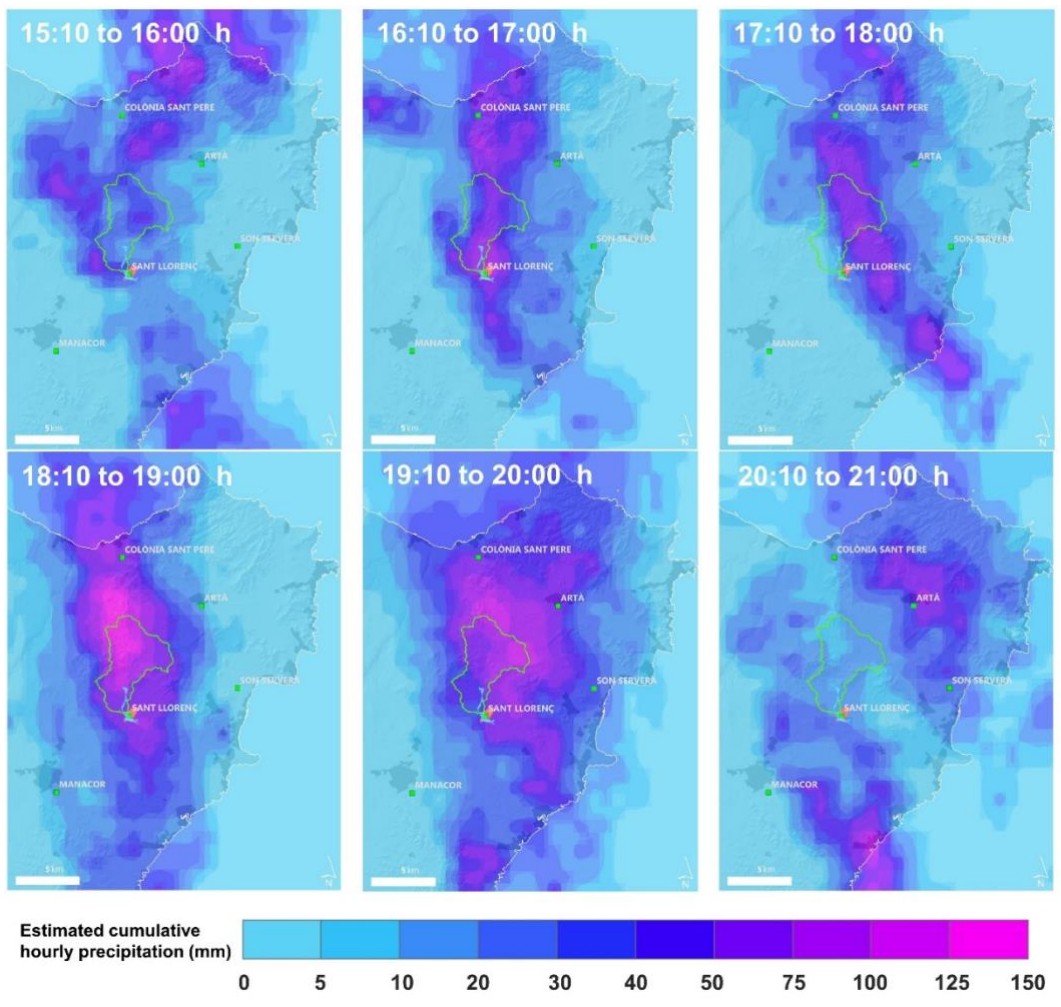

**Figure 4**. Evolution of the spatial distribution of the most intense hourly accumulated precipitations (i.e. ≥ 25 mm) over eastern Mallorca between 1600 h and 2100 h LT (1400 h and 1900 h UTC) on 9 October 2018.

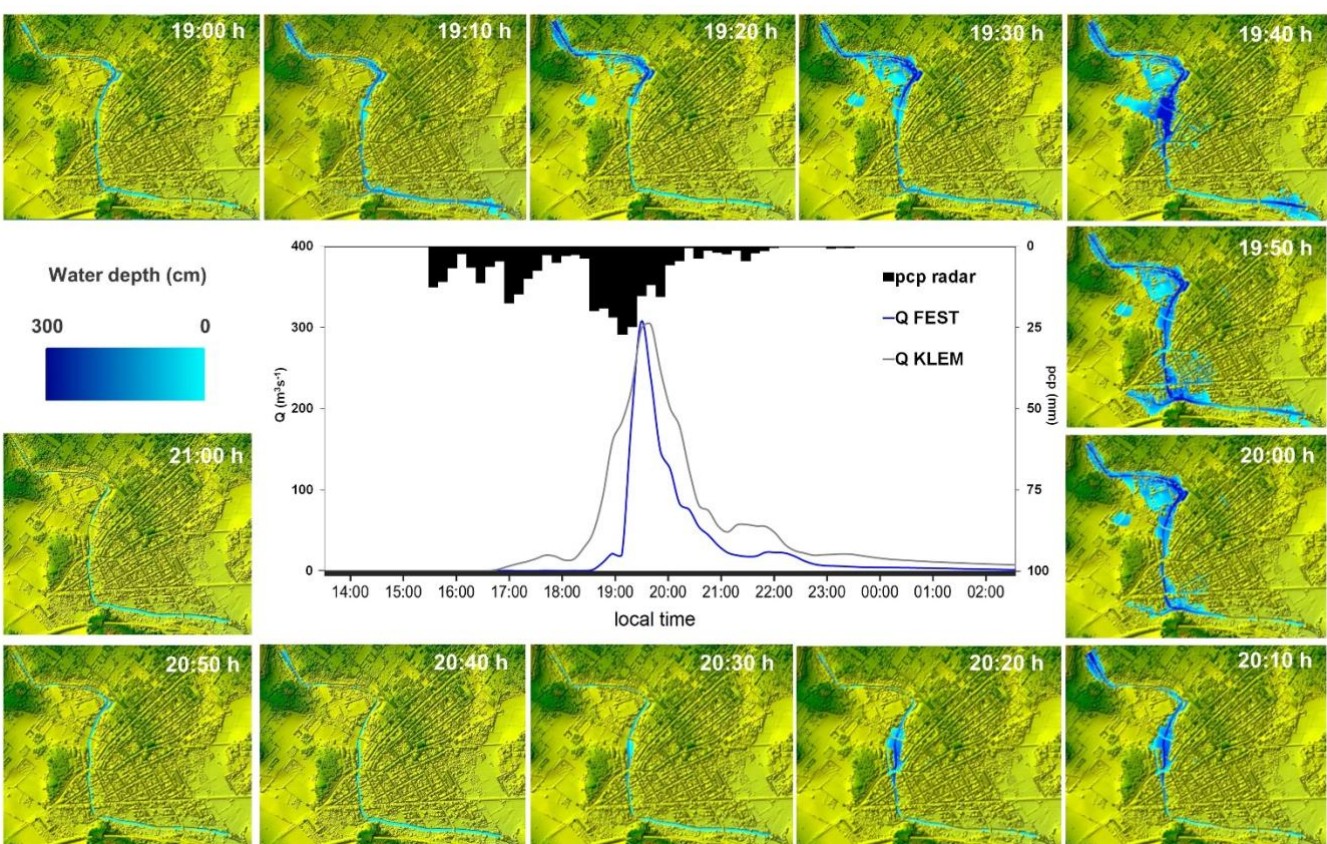

**Figure 5.** Modelled flash flood hydrographs at the entrance of Sant Llorenç and simulation of the flood peak as it crosses, devastating the town centre between 1900 h and 2100 h.

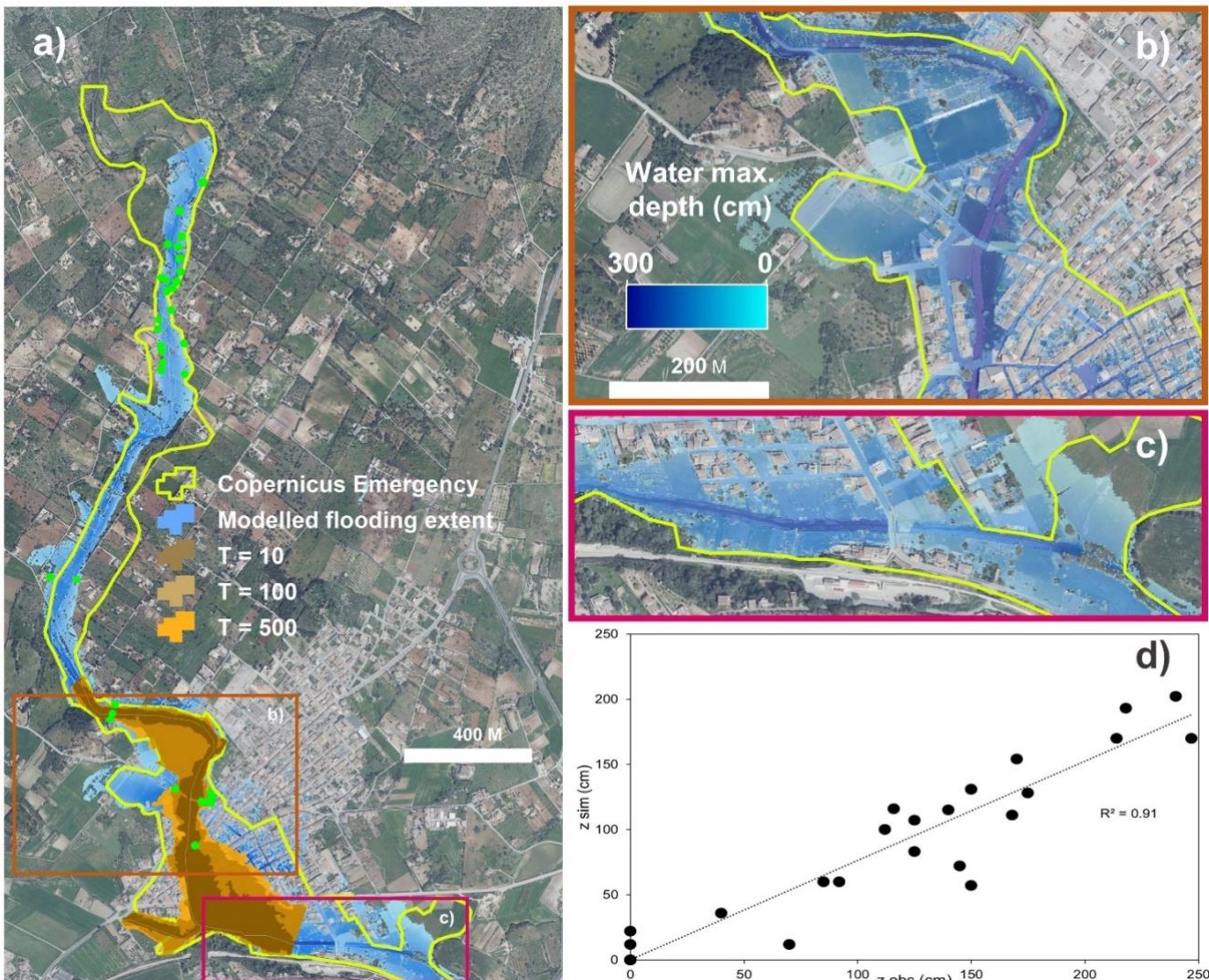

**Figure 6**. (a) Simulated flooding extent on the village (blue) compared with the observed event by Copernicus Emergency
Management System (yellow contour) and selected return periods (T=10, T=100, T=500); green dots depict the flooding marks
taken as reference to validate the simulation. (b and c) Detailed views of modelled maximum water depths at Sant Llorenç
town. (d) Scatter plot showing the relationship between simulated water depths and in-situ measured flooding marks.

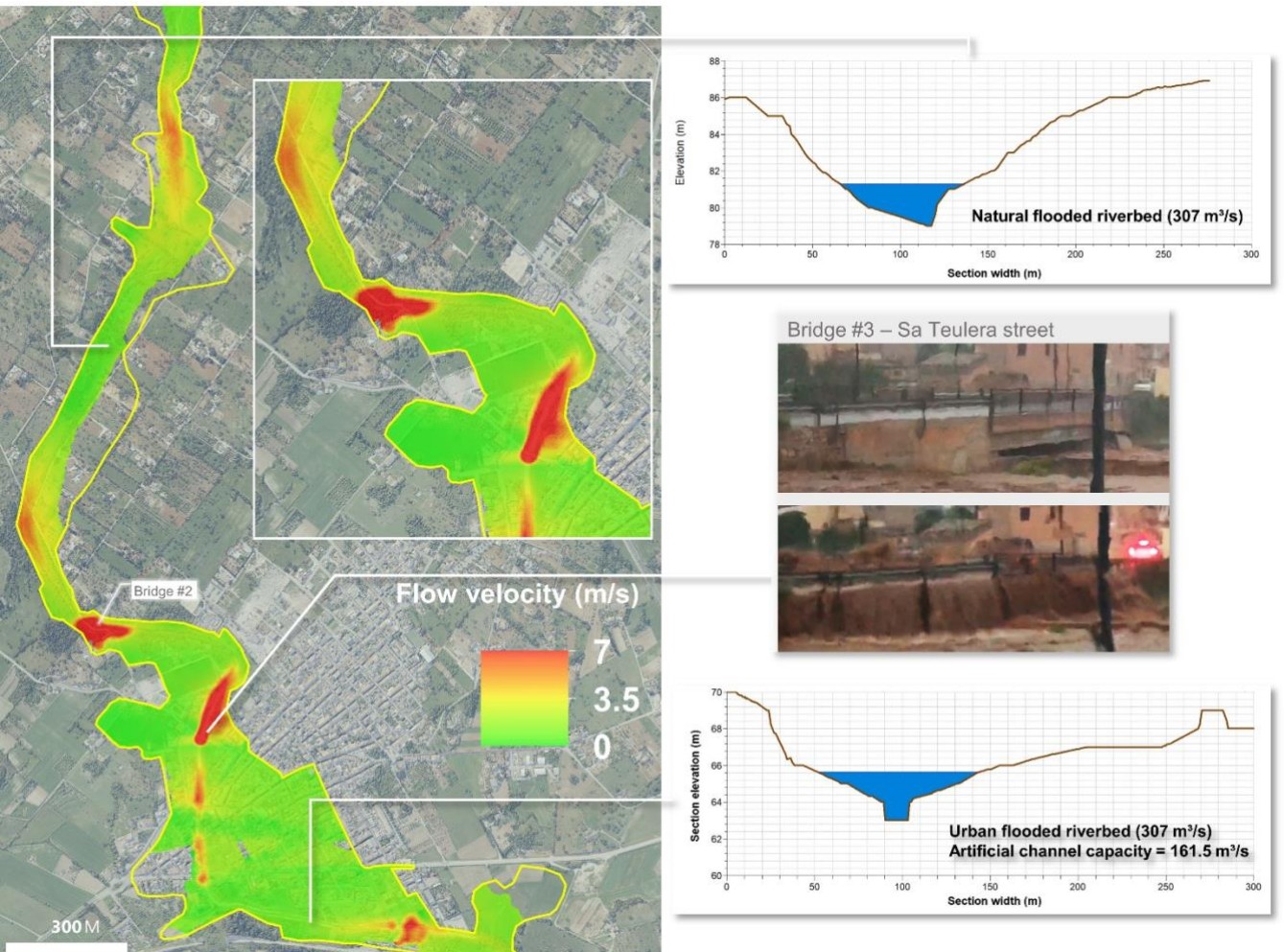

**Figure 7.** Left panel: simulated maximum flow velocity across Sant Llorenç town during the event. Yellow contour line depicts the observed event by Copernicus Emergency Management System. Right panel: maximum discharge flowing through two sections, calculated from flooding marks and LIDAR point clouds, including velocities on the artificial concrete channel.

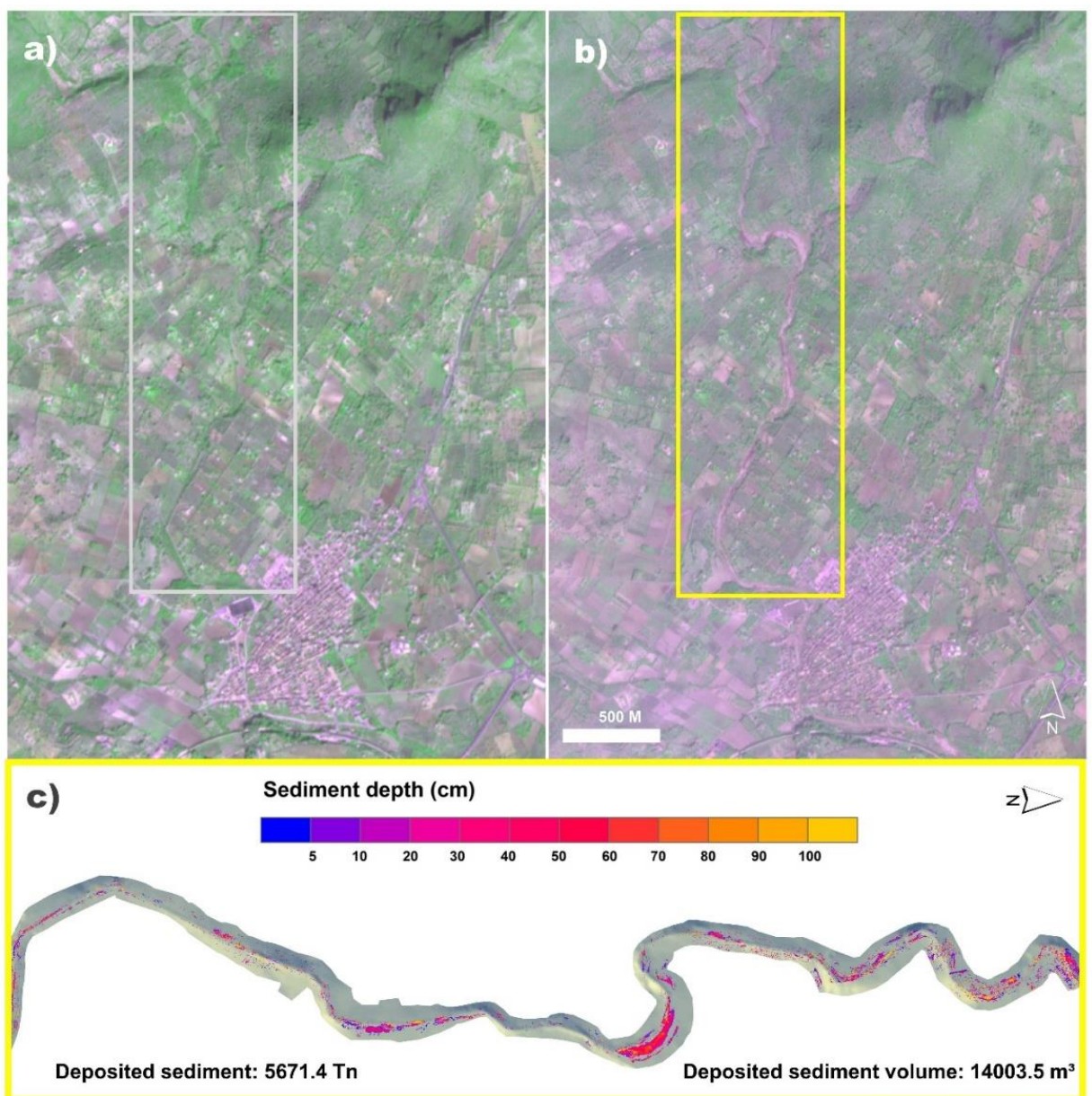

**Figure 8.** Effects of the catastrophic flash flood on the agricultural surroundings of Sant Llorenç town: sharpened false infrared RGB composites before, on October 9 (a), and after the event, on October 11 (b), derived from Planet® high resolution (3 m) imagery; (c) spatial patterns of sediment deposition and accumulation.

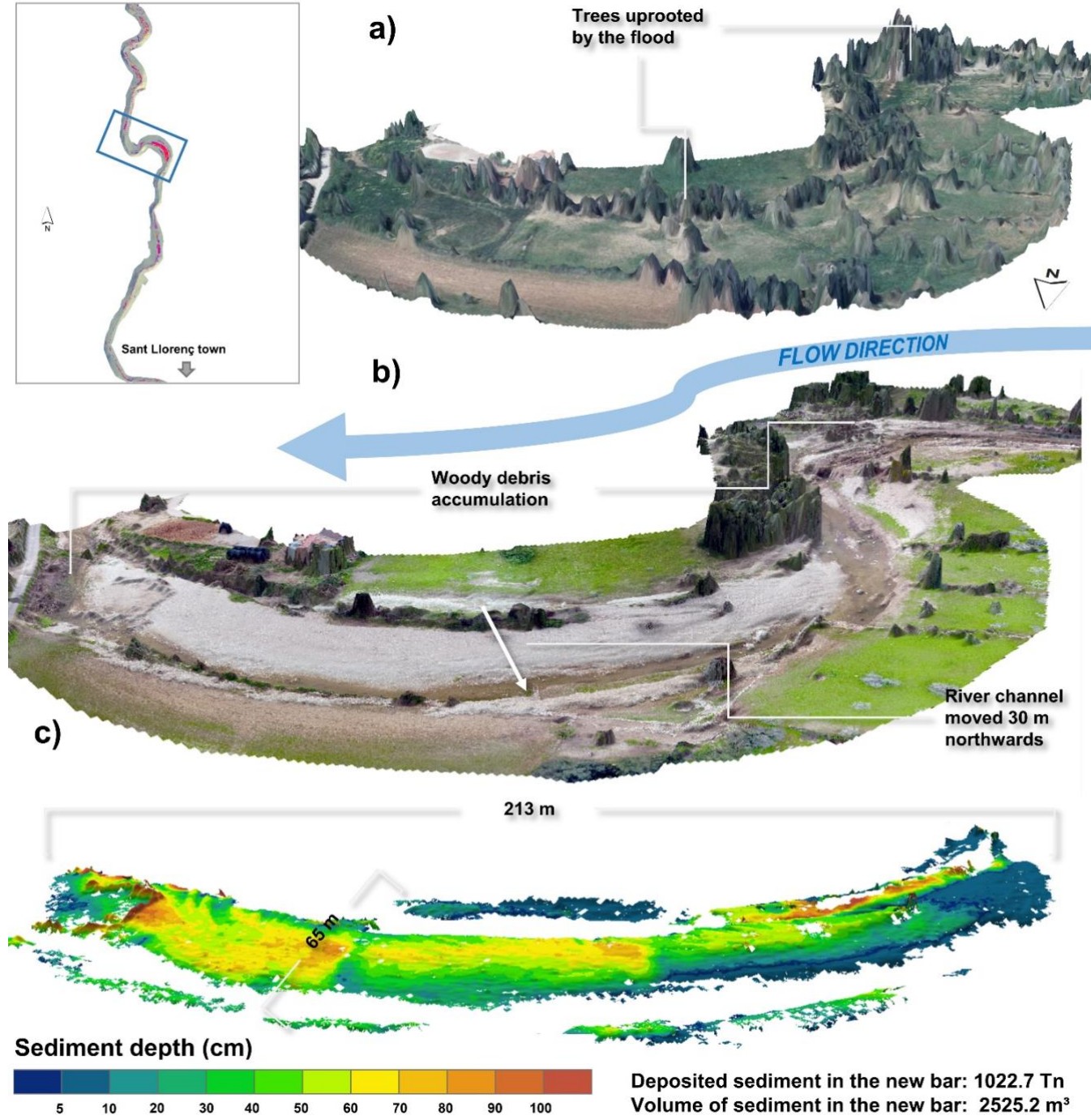

**Figure 9.** View of the Ses Planes stream at meander 2 km upstream of Sant Llorenç town: (a) December 2014 orthophotograph
(25 cm) superimposed onto a LIDAR-derived DEM (1m) before the event, and (b) November 2018 orthophotograph (3 cm)

dropped onto an SfM-derived DEM (6 cm) after the flash flood. (c) Geomorphological changes and details of the sediment bar created during the flash flood at the meander.