# Peer review of "Hydro-meteorological reconstruction and geomorphological impact assessment of the October, 2018 catastrophic flash flood at Sant Llorenç, Mallorca (Spain)"

_Natural Hazards and Earth System Sciences, 2019_

## Referee Comment (RC1) · Anonymous Referee #1 · 15 Aug 2019

**Review of the paper "Hydro-meteorological reconstruction and geomorphological impact assessment of the October, 2018 catastrophic flash flood at Sant Llorenç, Mallorca (Spain) submitted to NHESS by Jorge Lorenzo-Lacruz, Celso Garcia, Enrique Morán-Tejeda, Arnau Amengual, Víctor Homar, Aina Maimó-Far, Alejandro Hermoso, Climent Ramis, Romualdo Romero**

**General Comments**

The paper focus on the hydrometeorological diagnostic of a catastrophic flash flood event that affected a small semi-arid and torrential basin in Mallorca on October 2018. Rainfall evolution has been estimated from radar data, while the discharge has been obtained applying FEST-WB and the hydraulic simulation has been done through HEC-RAS.  of the event and mapping of affected areas.

It is an interesting case of study and a deep study that merits be published. The main problem is the hydrological model that the authors have used, that is not the more adequate to be applied in flash floods events in little catchments that the case of study. Authors should justify better the suitability of this model.

Besides other changes that I expose a continuation, the entire paper needs a careful revision of the English language by a native and official translator. Some paragraphs of the present version have great difficulties to be understood.

**Abstract:**

Besides the language revision, some information should be added in the abstract: the date in which the event was produced, the hydrological model, the time step used in this model, the hydraulic model, the spatial resolution for modelling the flooded area. They should indicate the data source. Authors also say that the flooded area exceeded the extension for a 500-year return period flood, but they should indicate the source of this flood hazard map that they use for the comparison.

**Introduction**

The Introduction doesn't reflect the deep knowledge on heavy rainfalls and floods in Mediterranean Areas of some of the co-authors.  It should be improved, both from the meteorological point of view and hydrological point of view. It offers a poor and non-updated knowledge of the state of the art about heavy precipitation events and flash floods in Mediterranean Areas, while the authors are experts on the matter. The own authors have important contributions on this matter that would be useful for this paper. For instance, in this paper, authors state the importance of the intrusion of cold polar air masses aloft, when this factor is not present in the major part of the cases. On the contrary, they do not say anything about the important role played by mesoscale or synoptic lows.

Besides this they do not say anything about the hydrological approaches that are usually applied to this kind of events, mainly if we consider the transport of solid material that can affect considerably the estimation of the discharge. This is an important problem when they decide to use a non-usual hydrological model.

As example of literature to improve this Introduction I recommend (besides other publications of some of the co-authors):

Martínez, C. et al, 2008. Heavy rain events in the Western Mediterranean: an atmospheric pattern classification. Adv. Sci. Res., 2, 61–64, 2008 www.adv-sci-res.net/2/61/2008/

Lumbroso, D., Gaume E., 2012. Reducing the uncertainty in indirect estimates of extreme flash flood discharges, Journal of Hydrology, doi:10.1016/j.jhydrol.2011.08.048

Ducrocq V. et al., 2014. HYMEX-SOPI The Field Campaign Dedicated to Heavy Precipitation and Flash Flooding in the Northwestern Mediterranean. Bulletin of the American Meteorological Society, 95(7): 1083.

Gaume, E. et al, 2016. Mediterranean extreme floods and flash floods. Into Hydro-meteorological extremes, chapter 3, The Mediterranean Region under Climate Change. A Scientific Update (coordinated byAllEnvi).133-144. IRD Éditions Institut de Recherche pour le Développement, Marseille, 2016, ISBN : 978-2-7099-2219-7

Llasat, M.C. et al., 2016. Trends in flash flood events versus convective precipitation in the mediterranean region: the case of catalonia. Journal of Hydrology, 541, 24-37, http://dx.doi.org/10.1016/j.jhydrol.2016.05.040 0022-1694

Hally et al, 2015. Hydrometeorological multi-model ensemble simulations of the 4 November 2011 flash flood event in Genoa, Italy, in the framework of the DRIHM project. Nat. Hazards Earth Syst. Sci., 15, 537–555, 2015

**Catchment description:**

Add more information about the radar (temporal and spatial resolution, number of vertical scans, products that you have used). Clarify if you have worked with the raw echo radar imagery provided by AEMET without any correction of if you have worked with corrected images or precipitation products provided by AEMET.

**Atmospheric modelling and convective precipitation predictability**

The statement "An initial numerical exploratory study was performed after it was ascertained that no operational system forecasted 135 precipitation rates over eastern Mallorca anywhere near the recorded rainfall rates (Figs. 2c and 2d) in their operational cycles" is a very serious denounce that is not enough justified. As instance, authors mention Figure 2 to justify this accusation but Figure 2 shows results from WRF when the forecast made by AEMET was with another model. If you want to maintain this statement you need to be stricter and show all the operational models that you are referring.

In line 147 you speak again about "the WRF runs were nested in the 00 UTC October 9 operational cycle", but which operational cycle?

All the paragraphs where authors present the methodology to modify the WRF simulation should be moved to **Methodology.** It will be better to move the entire section 2.3 to Methodology

**Hydrological modelling**

The Flood Event–Based Spatially Distributed Rainfall–Runoff Transformation–Water Balance (FEST-WB) model is not adequate for flash-flood events. Evapotranspiration do not play any role in these cases. Authors justify it by the reference by Rabuffetty but in this case it was applied to

Po River, that has a major catchment. Please, look for more references on the use of this model in cases of flash-floods. The better models for this kind of event are DRiFT, RIBS or HBV.

**Results**

Authors state that radar-derived rainfall estimates showed very high agreement with rain gauge data. Usually radar data products are calibrated and corrected with measures in surface provided by raingauge networks. Do the authors know if they had work with the **** radar products or corrected products? They should clarify this in the paper.

Authors say that hydraulic simulation showed that water reached a depth of 3 m at some points, and modelled water depths highly correlate ($R^2 = 0.91$) with in-situ after-event measurements. They should indicate the number of measures and location.

Authors indicate that the flash flood eroded and transported woody and abundant sediment debris, changing channel geomorphology. How had they considered this transport in the hydrological simulation? I would recommend them to read the paper by Martin Vide and Llasat (2018) were this kind of problem is analysed in detail for another flash flood event that was produced in a neighbouring region.

> Martín-Vide, J.P., M.C. Llasat, 2018.    The 1962 flash flood in the Rubí stream river (Barcelona, Spain). Journal of Hydrology 566, 441–454.

**Discussion**

Authors say that the development of a successful flash flood warning system for the region will require of a hydro-meteorological forecasting system that combine sub-kilometric precision in the precipitating systems, decametric precision in the hydrological modelling and metre or sub-metre precision in the hydraulic component of the forecasting chain. I am afraid that this last part related with this precision in hydrological modelling is not needed and non-realistic. For this case of events, where heavy precipitation plays the most important role in a catchment with high flood risk, an improvement of QPF by using blending techniques plus the improvement of the mesoscale models and radar nowcasting will provide a good advancement.

**Minor changes:**

Lines 14-16. The text is the same that this one of lines 55-58. Please, modify. Besides this, it doesn't contain all the required information

Line 30. Please, indicate for which period had Spain reported more than 20 floods per 10,000 km2, with 652 fatalities.

Line 34. Please, indicate the dates and regions in which flash floods were produced between October 9 and November 9, 2018.

Line 40. The greatest part of heavy precipitation events that produced flash floods in Mediterranean Region are not related with the intrusion of cold polar air masses aloft. Please, don't include this condition in the new Introduction.

Line 43. Authors say that "High precipitation rates can remain during several hours over individual catchments." This is correct but it should be important to clarify that some important flash floods are produced by heavy precipitations that last less than 1 hour

Line 61. Please, substitute "Meteorology-based prediction methods" by "the operative mesoscale meteorological model".

Line 78. Add a reference to justify this sentence about Hortonian flows during intense rainfall episodes.

Line 85. "In this study,…" About which study are the authors speaking?

Lines 134-135. Indicates to which operational system are you referring (mesoscale model, resolution, data provider)

Line 148. Substitute Oct by October

Line 178. The sentence "distribution of accumulated radar-derived precipitation (Fig. 2c and 3a)" is not true. Figure 2c provides WRF results.

Line 185. What is the meaning here of "land mass"?

Line 186. The expression "pluviometric density of 29.4 km²" is not correct. Modify it or modify the units

Line 339. Synoptic conditions are not the responsible of the stationarity of the convective system that generated a succession of convective nuclei.

**Figures**

Figure 1. Some legends of Figure 1 are not enough clear to be reproduced. Add the location of the radar.

Figure 2. Indicate the meaning of CC BY-NC) write in the text. Add to which WRF Model are you referring and the resolution, as well as the provider of these images

Figure 3. Add a radar imagery showing the structures that affected the catchment. Correct " 2018 flashflood."

---

## Referee Comment (RC2) · Anonymous Referee #2 · 26 Sep 2019

The paper presents a very good characterization of a very heavy rainfall event in Mallorca. Despite this is an isolated event, this a very good example on how analyzing this type of hazards from the meteorological origin, to the derived impacts of the flood. Such chain of analyses can serve as example to analyze a very common hazard affecting to many Mediterranean sites. The used methodology is robust and the authors provide evidence that it worked in a very reasonable way (despite the difficulty to get reliable observations for calibration and validation under such extreme conditions). The graphical material provided in the article is of great quality.

[Figure]

I have only minor comments in the structure of the article that authors may consider to prepare a definitive version of the manuscript. I think that the presentation of the synoptic characteristics of the storm should be moved to results section as they are presenting specific analyses for this event. In addition many other statements given when presenting results should probably be moved to discussion as they are not results themselves but they are hypothesis trying to explain the obtained results.

In addition the authors might consider to present how the weather forecast from the main meteorological models evolved the days before to predict the precipitation over this area. Authors said that models clearly underestimated the event, but it could be good to see more on that to illustrate to which extent the early warning systems (not real time ones) may work in these areas.

There are small typos over the text, so may be good a last slow reading to correct them.

---

## Author Comment (AC1) · 18 Oct 2019

**General Comments**

**The paper focus on the hydrometeorological diagnostic of a catastrophic flash flood event that affected a small semi-arid and torrential basin in Mallorca on October 2018. Rainfall evolution has been estimated from radar data, while the discharge has been obtained applying FEST-WB and the hydraulic simulation has been done through HEC-RAS. of the event and mapping of affected areas. It is an interesting case of study and a deep study that merits be published.**

We highly appreciated the constructive comments made by the reviewer, all of them aimed to improve the quality of the manuscript. Following we provide a list of answers (normal font) to reviewer's comments (bold), and the changes (between quotation) made in the manuscript. We also provide a marked changes version of the manuscript for a correct tracking of the changes included in this new version.

**The main problem is the hydrological model that the authors have used, that is not the more adequate to be applied in flash floods events in little catchments that the case of study. Authors should justify better the suitability of this model.**

The FEST model is a fully-distributed physically-based continuous hydrological model that relies in solving the water balance equation. The FEST has been developed with the aim of flexibility and it can be applied to a wide range of spatial and temporal scales and climatic conditions. If necessary, the water balance equation can be solved at the point scale. Particularly, FEST has been used not only over large catchments in northern Italy, but over small-to-medium size basins in north and south Italy and eastern Spain. In addition, it has been successfully implemented over catchments in China or Brazil, for instance. Some of these FEST applications are referenced in the manuscript (e.g. Montaldo et al, 2007; Ravazzani et al., 2016; Amengual et al., 2017).

Nevertheless, a second hydrological model has been implemented within Ses Planes catchment up to Sant Llorenç town, so as to cope with reviewer's concerns about the use of FEST model over a small and semi-arid basin. The authors have implemented the KLEM model. KLEM is a simple fully-distributed kinematic event-based hydrological model widely used in flash flood research and for assessing flood impacts. Originally, KLEM was designed to simulate flash floods over small and natural mountainous catchments similar to the Ses Planes basin, although over time it has also been applied to a wide range of basin sizes. Some notable papers that derived their findings after implementation of the KLEM model are:

Giannoni, F., J. A. Smith, Y. Zhang, and G. Roth, 2003: Hydrologic modeling of extreme floods using radar rainfall estimates. Adv. Water Resour., 26, 195–200.

Borga M, Boscolo P, Zanon F, Sangati M (2007) Hydrometeorological analysis of the 29 August 2003 flash flood in the Eastern Italian Alps. J Hydrometeor 8(5): 1049-1067.

Zoccatelli, D., et al., 2010. Which rainfall spatial information for flash flood response modelling? A numerical investigation based on data from the Carpathian range, Romania. Journal of Hydrology, 394 (1–2), 148–161. doi:10.1016/j. jhydrol.2010.07.019.

**Besides other changes that I expose a continuation, the entire paper needs a careful revision of the English language by a native and official translator. Some paragraphs of the present version have great difficulties to be understood.**

The first submitted manuscript already undergone an English editing process by a hired native speaker. See link below:

http://www.mpomeroyenglishediting.ca/

Nevertheless, we will check again English language and grammar in order to correct misspellings and typos.

**Abstract: Besides the language revision, some information should be added in the abstract: the date in which the event was produced, the hydrological model, the time step used in this model, the hydraulic model, the spatial resolution for modelling the flooded area. They should indicate the data source. Authors also say that the flooded area exceeded the extension for a 500-year return period flood, but they should indicate the source of this flood hazard map that they use for the comparison.**

Some of the information suggested by the reviewer was already included in the abstract (date of the event). We added some of the information suggested that was missed (models used). However, in order to be concise and respect abstract word limits, the following information, more technical, was only included in the methods section, and not in the abstract: time step, spatial resolution of flood mapping and source of the flood return periods maps.

**Introduction**

**The Introduction doesn't reflect the deep knowledge on heavy rainfalls and floods in Mediterranean Areas of some of the co-authors. It should be improved, both from the meteorological point of view and hydrological point of view. It offers a poor and non-updated knowledge of the state of the art about heavy precipitation events and flash floods in Mediterranean Areas, while the authors are experts on the matter. The own authors have important contributions on this matter that would be useful for this paper. For instance, in this paper, authors state the importance of the intrusion of cold polar air masses aloft, when this factor is not present in the major part of the cases. On the contrary, they do not say anything about the important role played by mesoscale or synoptic lows. Besides this they do not say anything about the hydrological approaches that are usually applied to this kind of events, mainly if we consider the transport of solid material that can affect considerably the estimation of the discharge. This is an important problem when they decide to use a non-usual hydrological model. As example of literature to improve this Introduction I recommend (besides other publications of some of the co-authors):**

**Lumbroso, D., Gaume E., 2012. Reducing the uncertainty in indirect estimates of extreme flash flood discharges, Journal of Hydrology, doi:10.1016/j.jhydrol.2011.08.048**

**Ducrocq V. et al., 2014. HYMEX-SOPI The Field Campaign Dedicated to Heavy Precipitation and Flash Flooding in the Northwestern Mediterranean. Bulletin of the American Meteorological Society, 95(7): 1083.**

**Gaume, E. et al, 2016. Mediterranean extreme floods and flash floods. Into Hydro-meteorological extremes, chapter 3, The Mediterranean Region under Climate Change. A Scientific Update (coordinated byAllEnvi).133-144. IRD Éditions Institut de Recherche pour le Développement, Marseille, 2016, ISBN : 978-2-7099-2219-7**

**Llasat, M.C. et al., 2016. Trends in flash flood events versus convective precipitation in the mediterranean region: the case of Catalonia. Journal of Hydrology, 541, 24-37, http://dx.doi.org/10.1016/j.jhydrol.2016.05.040 0022-1694**

**Hally et al, 2015. Hydrometeorological multi-model ensemble simulations of the 4 November 2011 flash flood event in Genoa, Italy, in the framework of the DRIHM project. Nat. Hazards Earth Syst. Sci., 15, 537–555, 2015**

We appreciate the reviewer's comments and agree that this event is a very good example of the general problematic of flash floods in the western Mediterranean region. Note that this study is mainly focused

on the hydrologic-hydraulic aspects of the Sant Llorenç event. The Introduction aims at providing a brief description of the main synoptic and mesoscale mechanisms driving this type of events, rather than a general review of meteorological aspects of Mediterranean flash floods. In addition, section 2.2 is fully devoted to describing the large-scale ingredients linked to this particular case and also to heavier precipitations of convective origin affecting Mediterranean Spain. Nevertheless, we improved key arguments in the Introduction and added a few more references:

Martínez, C. et al, 2008. Heavy rain events in the Western Mediterranean: an atmospheric pattern classification. Adv. Sci. Res., 2, 61–64, 2008 www.adv-sci-res.net/2/61/2008/

Since the authors use a physically-based hydrological model to derive the main results of this work, there is no need of using simplifications or approximations when simulating this extreme flash flood. Essentially, the model solves the physical laws that have been derived for the different hydrological processes. Sub-grid processes are parameterized by using empirical values derived after a long research in the field of physical hydrology. These equations are solved by standard and well-tested numerical schemes not just employed in the field of physical hydrology, but in the wide field of solving numerically differential equations in several scientific disciplines.

The authors would highlight again that the FEST model is not a "non-usual hydrological model". As aforementioned, FEST has been successfully applied in many basins around the world. The equations implemented in FEST are taken from state of the art literature. Moreover, further distributed models have been derived and from FEST, such as: AFFDEF (Moretti and Montanari, 2007) and DIMOSOP (Ranzi et al., 2003).

The 9 October 2018 flash flood was not a hyperconcentrated debris flow. After recollecting information from the in-situ post-event field work, the authors estimated a total sediment volume of $1.4 \cdot 10^{-2}$ Hm3. Hydrological simulations yield a total water volume ranging from 1.0 to 1.9 Hm3. Therefore, sediment concentration (0.7-1.4%) was well below 6% of the total volume, safely disregarding a debris flow (Pierson, 2005). During the 9 October 2018 flash flood, most sediment was transported as bed load and no appreciable impacts on the simulated peak discharges or timings can be expected. This fact was confirmed by examining real-time images and videos taken during the flash-flood. In addition, peak discharges, timings and flow velocities have not been derived from the hydrological model's simulations per se, but they have been estimated from meticulous in-situ post-event field measurements along the Ses Planes ephemeral stream. The hydrological models have been calibrated from these estimations and not the opposite.

In light of the numerous concerns shown by the reviewer, some confusing statements in the original manuscript have been further clarified in subsections 3.2, 4.2 and 4.4, and additional information and explanations have been added in the revised version of the manuscript.

Moretti, G., and Montanari, A. (2007). "AFFDEF: A spatially distributed grid based rainfall-runoff model for continuous time simulations of river discharge." Environ. Modell. Software, 22(6), 823–836.

R. Ranzi, B. Bacchi, and G. Grossi (2003). Runoff measurements and hydrological modelling for the estimation of rainfall volumes in an alpine basin. Quart. J. Royal Meteor. Soc., 129: 653-672.

**Catchment description: Add more information about the radar (temporal and spatial resolution, number of vertical scans, products that you have used). Clarify if you have worked with the raw echo radar imagery provided by AEMET without any correction of if you have worked with corrected images or precipitation products provided by AEMET.**

Done. This information has been included in the revised subsection 3.1.

**Atmospheric modelling and convective precipitation predictability The statement "An initial numerical exploratory study was performed after it was ascertained that no operational system forecasted 135 precipitation rates over eastern Mallorca anywhere near the recorded rainfall rates (Figs. 2c and 2d) in their operational cycles" is a very serious denounce that is not enough justified. As instance, authors mention Figure 2 to justify this accusation but Figure 2 shows results from WRF when the forecast made by AEMET was with another model. If you want to maintain this statement you need to be stricter and show all the operational models that you are referring.**

The beginning of section 2.3 has been extended in order to provide further details on the severe underforecast of this event by the official operational models. In particular, we mention the systematic underestimations by all cycles of ECMWF and HARMONIE-AROME. As a graphical example, we show below the deterministic ECMWF forecast rainfall at 3-h intervals provided by the cycle starting at 9 October 00 UTC. We believe adding this kind of figures in the manuscript is unnecessary given the hydrologic-hydraulic main orientation of the study.

[Figure]

**In line 147 you speak again about "the WRF runs were nested in the 00 UTC October 9 operational cycle", but which operational cycle?**

We are referring to the ECMWF EPS operational cycle. A clarification of this point has been added to the text (see marked changed version).

**All the paragraphs where authors present the methodology to modify the WRF simulation should be moved to Methodology. It will be better to move the entire section 2.3 to Methodology**

Following the recommendations of the reviewer, we moved the first half of section 2.3 to the Methodology; the second half of the section, which includes results about the predictability of the precipitation event, was moved to a new sub-section within the Results section. In this fashion, we believe that the paper is much better organized.

**Hydrological modelling The Flood Event–Based Spatially Distributed Rainfall–Runoff Transformation–Water Balance (FEST-WB) model is not adequate for flash-flood events. Evapotranspiration do not play any role in these cases. Authors justify it by the reference by Rabuffetty but in this case it was applied to Po River, that has a major catchment. Please, look**

**for more references on the use of this model in cases of flash-floods. The better models for this kind of event are DRiFT, RIBS or HBV.**

The authors are well aware that evapotranspiration does not play any substantial role in water abstractions when dealing with a flash flood, as the temporal scale of this natural hazard is very short. However, the FEST model relies on a realistic initialization of soil moisture. The initial soil moisture is used to update the maximum potential retention in the SCS-CN method. Note that the role of the initial soil moisture cannot be neglected on the flood response of extreme events, especially when combined with high soil moisture capacities. Both factors have a noticeable impact in terms of enhanced nonlinearities related to the wetting-up processes and mitigation of flood peak magnitude with respect to rainfall magnitude. See Borga et al. (2007) and Amengual et al. (2017) for further information about these issues. As FEST solves the water balance equation, the evapotranspiration is relevant when computing the water balance used to assess the initial soil moisture content. However, during the simulation of a flash flood, FEST does not subtract water via evapotranspiration. That is, the evapotranspiration is computed at daily scale so as to close the water balance equation. Further clarifications respect this issue has been explicitly included in the revised version of the paper to avoid further confusions about how the model deals with evapotranspiration.

The authors appreciate reviewer's suggestion about the use of DRIFT, RIBS or HBV when simulating flash flood over small and semi-arid catchments. Even if they are not too familiarized with the use of these hydrologic models, two of them appear to be conceptual and lumped or semi-distributed. In this case study, as the authors have employed high-resolution spatial and temporal precipitation fields derived from radar observations, it is more advisable to use fully-spatially distributed hydrological models so as to simulate more realistically the interplay among the rainfall fields, basin physiography, river network, and soil properties. By using lumped or semi-distributed conceptual models, some theoretical assumptions must be done and the spatial-averaged features of the aforementioned fields must be employed.

**Results**

**Authors state that radar-derived rainfall estimates showed very high agreement with rain gauge data. Usually radar data products are calibrated and corrected with measures in surface provided by raingauge networks. Do the authors know if they had work with the **** radar products or corrected products? They should clarify this in the paper.**

This information has been included in the revised subsection 3.1.

**Authors say that hydraulic simulation showed that water reached a depth of 3 m at some points, and modelled water depths highly correlate ($R^2 = 0.91$) with in-situ after-event measurements. They should indicate the number of measures and location.**

We added the number of post-flood measures (32) in-text at the end of section 3.3. Flooding marks locations are already depicted in Figure 6a. We changed the symbology (color) of the flooding marks to increase its visibility.

**Authors indicate that the flash flood eroded and transported woody and abundant sediment debris, changing channel geomorphology. How had they considered this transport in the hydrological simulation? I would recommend them to read the paper by Martin Vide and Llasat (2018) were this kind of problem is analysed in detail for another flash flood event that was produced in a neighbouring region.**

**Martín-Vide, J.P., M.C. Llasat, 2018. The 1962 flash flood in the Rubí stream river (Barcelona, Spain). Journal of Hydrology 566, 441–454.**

As aforementioned, the 9 October 2018 flash flood was not a hyperconcentrated flow or debris flow. Sediment transported had concentration less than the critical threshold. Sediment concentration was at most 1.4% of the total volume. Therefore, impacts on water flood behavior can be safely neglected.

**Discussion**

**Authors say that the development of a successful flash flood warning system for the region will require of a hydro-meteorological forecasting system that combine sub-kilometric precision in the precipitating systems, decametric precision in the hydrological modelling and metre or submetre precision in the hydraulic component of the forecasting chain. I am afraid that this last part related with this precision in hydrological modelling is not needed and non-realistic. For this case of events, where heavy precipitation plays the most important role in a catchment with high flood risk, an improvement of QPF by using blending techniques plus the improvement of the mesoscale models and radar nowcasting will provide a good advancement.**

The full paragraph has been rewritten to better express the challenges and requirements for sub-urban quantitative hydraulic forecasts (see marked changes version).

**Minor changes:**

**Lines 14-16. The text is the same that this one of lines 55-58. Please, modify. Besides this, it doesn't contain all the required information**

Text in lines 55 to 58 has been modified to avoid repetition, as follows:

"The methodology used for the reconstruction of the event was organized on three main stages: (i) 10-minutes precipitation has been derived from radar reflectivity observations; (ii) two distinct fully-distributed hydrological models have been implemented to accurately simulate the discharge hydrograph and; (iii) a hydraulic simulation has been performed to map the affected areas, including flooding extent and timing, water depth and water velocity. Some of the geomorphological impacts on the main channel have been also assessed by using very high-resolution orthophotographs and digital elevations models for comparison of pre- and post-flooding conditions."

All the required information is contained in the methods section, and showed in a more detailed way. Here, in the introduction, we only show a brief overview of the methodology used.

**Line 30. Please, indicate for which period had Spain reported more than 20 floods per 10,000 km2, with 652 fatalities.**

The information is on the first sentence of the paragraph (1980-2015), therefore the report is referred to 35 years.

**Line 34. Please, indicate the dates and regions in which flash floods were produced between October 9 and November 9, 2018.**

The following text was added:

"The deadliest flash floods occurred in the Mediterranean area (Mallorca, Tunisia, Veneto, Sicilia, and Jordan), including 5 events (9, 17 and 29th October; 3 and 9th November) with 70 fatalities".

**Line 40. The greatest part of heavy precipitation events that produced flash floods in Mediterranean Region are not related with the intrusion of cold polar air masses aloft. Please, don't include this condition in the new Introduction.**

The term "polar mass" has been removed from the introduction

**Line 43. Authors say that "High precipitation rates can remain during several hours over individual catchments." This is correct but it should be important to clarify that some important flash floods are produced by heavy precipitations that last less than 1 hour**

The whole paragraph has been rewritten following the recommendations of the reviewer (see marked changes version of the manuscript).

**Line 61. Please, substitute "Meteorology-based prediction methods" by "the operative mesoscale meteorological model".**

Done.

**Line 78. Add a reference to justify this sentence about Hortonian flows during intense rainfall episodes.**

We added the following reference:

Estrany, J., Garcia, C. and Alberich, R.: Streamflow dynamics in a Mediterranean temporary river, Hydrol. Sci. J., 55(5), 717–736, doi:10.1080/02626667.2010.493740, 2010.

**Line 85. "In this study,…" About which study are the authors speaking?**

We replaced "in this study" by "in the present study" to clarify this point.

**Lines 134-135. Indicates to which operational system are you referring (mesoscale model, resolution, data provider)**

The full paragraph has been reworded, providing further details on these issues.

**Line 148. Substitute Oct by October Line 178.**

Done.

**The sentence "distribution of accumulated radar-derived precipitation (Fig. 2c and 3a)" is not true. Figure 2c provides WRF results.**

We do not understand this suggestion. Figure 2c shows the "Probability of 12h-accumulated precipitation exceeding 100 mm valid at 00 UTC October 10, 2018 from the larger WRF domain initialized at 00 UTC October 9", as stated in figure caption. Figure 3a shows the "Spatial distribution of the 24-h accumulated radar-estimated precipitation for the October 9, 2018 flash flood", not WRF results, as it is also depicted in figure caption.

**Line 185. What is the meaning here of "land mass"?**

The authors mean "for computation just over the land mass" that the sea domain of the study area has not been considered when comparing radar estimations against rain-gauge observations. The sentence has been rewritten in order to clarify this point.

**Line 186. The expression "pluviometric density of 29.4 km²" is not correct. Modify it or modify the units**

We rephrased the sentence as follows:

"This inland region has a whole extension of 618.0 km² and a pluviometric rain gauge density of 0.034 gauges  km-² (one pluviometric station per 29.4 km²)."

**Line 339. Synoptic conditions are not the responsible of the stationarity of the convective system that generated a succession of convective nuclei.**

We rephrased the sentence to clarify that the responsible for the precipitation event were the convective storms train and not the synoptic conditions:

"The aforementioned synoptic conditions generated a succession of convective nuclei over the area, that triggered this extreme precipitation event".

**Figures**

**Figure 1. Some legends of Figure 1 are not enough clear to be reproduced. Add the location of the radar.**

We resized the legends to favor a correct visualization and reproduction of the figure. Also note that this figure is conceived to be reproduced and printed at full-page size.

We added the location of the radar in the location map.

**Figure 2. Indicate the meaning of CC BY-NC) write in the text. Add to which WRF Model are you referring and the resolution, as well as the provider of these images**

The acronym CC BY-NC refers to Creative Commons non-commercial use permission and it is cited in figure caption as indicated at https://www.wetterzentrale.de.

**Figure 3. Add a radar imagery showing the structures that affected the catchment.**

Done. A new figure (Fig. 4) have been added in the revised version of the paper.

**Correct " 2018 flashflood.**

Done.

---

## Author Comment (AC2) · 18 Oct 2019

**The paper presents a very good characterization of a very heavy rainfall event in Mallorca. Despite this is an isolated event, this a very good example on how analyzing this type of hazards from the meteorological origin, to the derived impacts of the flood. Such chain of analyses can serve as example to analyze a very common hazard affecting to many Mediterranean sites. The used methodology is robust and the authors provide evidence that it worked in a very reasonable way (despite the difficulty to get reliable observations for calibration and validation under such extreme conditions). The graphical material provided in the article is of great quality.**

We kindly appreciated the reviewer opinion on our manuscript. Following we describe how we addressed the main changes suggested by the reviewer. For a better tracking of the corrections, note that our answers are shown in normal font and reviewer comments are shown in bold.

**I have only minor comments in the structure of the article that authors may consider to prepare a definitive version of the manuscript. I think that the presentation of the synoptic characteristics of the storm should be moved to results section as they are presenting specific analyses for this event.**

In this particular case, we do not agree with the reviewer. In section 2.2 (Synoptic situation), we do not show any of our analysis or original results. Here we introduce a description of the synoptic situation which triggered the catastrophic flash flood, and we based this description on the information provided by secondary sources. Moreover, we think that the provided description acts as an introductory element for the upcoming hydro-meteorological reconstruction of the event, which is the core of the manuscript. For these reasons, we believe this information belongs to the Case Study section, rather than to the Results section.

**In addition, many other statements given when presenting results should probably be moved to discussion as they are not results themselves but they are hypothesis trying to explain the obtained results.**

We agree with the reviewer, and consequently we moved some paragraphs from the results to the discussion (please see the marked changes version of the manuscript).

**In addition, the authors might consider to present how the weather forecast from the main meteorological models evolved the days before to predict the precipitation over this area.**

The first paragraph of section 2.3 has been modified, providing further details on these issues (please see marked changes version).

**Authors said that models clearly underestimated the event, but it could be good to see more on that to illustrate to which extent the early warning systems (not real time ones) may work in these areas.**

The beginning of section 2.3 has been extended in order to provide further details on the severe underforecast of this event by the official operational models. In particular, we mention the systematic underestimations by all cycles of ECMWF and HARMONIE-AROME. As a graphical example, we show below the deterministic ECMWF forecast rainfall at 3-h intervals provided by the cycle starting at 9 October 00 UTC. We believe adding this kind of figures in the manuscript is unnecessary given the hydrologic-hydraulic main orientation of the study.

[Figure]

**There are small typos over the text, so may be good a last slow reading to correct them.**

We carefully read the manuscript and corrected several typos detected (see marked version of the manuscript).

---

## Author Comment (AC4) · 18 Oct 2019

The comment was uploaded in the form of a supplement:
https://www.nat-hazards-earth-syst-sci-discuss.net/nhess-2019-226/nhess-2019-226-AC4-supplement.pdf